# Teaching LLMs When to Stop Seeking and Start Acting

## Abstract

Many tasks require machine learning models to strategically gather relevant information over multiple rounds of interaction before actually acting on a task. Strategic information gathering requires models to know not only how to effectively acquire information, but also when to stop gathering information and make a decision, in order to avoid overthinking or getting derailed when acting. In this paper, we formalize this problem and introduce Counterfactuals and Reasoning for Termination (CaRT), an approach for teaching LLMs when to stop seeking information. To appropriately learn when to terminate, CaRT fine-tunes LLMs using counterfactual pairs of trajectories, one where termination is appropriate and a minimally modified version of the same trajectory where it is not. It trains the LLM to explain the rationale for the termination decision in either case via verbal reasoning, and imbues this capability into the base LLM via fine-tuning. We instantiate CaRT in two domains: interactive medical diagnosis and math problem solving. In both domains, we find that CaRT improves the efficiency of information gathering and task success rate compared to other fine-tuning methods.

## 1 Introduction

Strategic information gathering is core to problem solving and decision making (Thrun, 1992; Nie et al., 2024). For example, when attempting to design and prescribe a course of treatment to a user, it is important to gather complete information about their symptoms. In many scenarios, information gathering relies not only on deciding how to acquire more information, but also on deciding *when the model has gathered enough information* to solve the task. A model that stops too late wastes resources (Fig. 1), while one that stops too early risks failure. Moreover, additional information can be detrimental in long-context cases, where it may lead the model to latch onto spurious positional information (Liu et al., 2023; Wang et al., 2023b). The ability to recognize

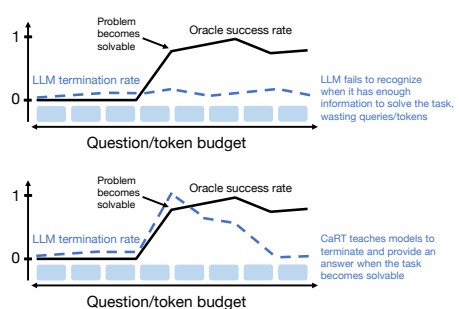

Figure 1: *A schematic illustration of termination behavior* with and without our proposed approach. While LLMs typically fail to recognize the best points to stop thinking or questioning, our approach **CaRT** imbues them with the ability to correctly identify this point.

when "I know enough" is therefore essential to efficient and reliable problem solving.

Deciding when to stop thinking or interacting, and hence stop seeking information, is difficult because it requires predicting the expected future utility of continuing under the model's current policy. Statistical approaches to this problem rely on accurately estimating a value function (Nie et al., 2019; Thomas and Brunskill, 2016) with limited data and typically operate in domains with well-defined environment dynamics, such as airline ticket purchasing (Groves and Gini, 2015; Goel et al., 2017). We introduce LLMs to this problem setting because they possess natural language capabilities and rich priors about the world, providing the potential for greater versatility, domain applicability, and generalization compared to statistical approaches. However, off-the-shelf LLMs even struggle to accurately predict their probability of success (Savage et al., 2024; Omar et al., 2025; Sun et al., 2025; Groot and Valdenegro-Toro, 2024) and are unable to conduct principled exploration (Arumugam and Griffiths, 2025; Tajwar et al., 2025). These limitations put into question

whether current recipes for training LLMs imbue them with the ability to quantify the value of what they don't know, a key skill for effective termination.

In this paper, we build an approach to imbue LLMs with the ability to stop or "terminate" their internal thinking processes and/or environment interaction at the right point for maximal performance, without wasting computation or interaction. Our key high-level insight is that reasoning itself can be used to learn accurate and generalizable termination behavior, as long as this reasoning is done comparatively (and contrasts the benefits of terminating and continuing). Our approach, **Counterfactuals and Reasoning for Termination (CaRT)**, fine-tunes models with *counterfactual* pairs: trajectories where termination is appropriate and minimally modified trajectories where it is not, combined with explicit natural language reasoning traces that justify why termination is the right decision. This comparative reasoning signal enables models to implicitly maintain a verbalized value function, allowing the LLM to foresee the benefits of termination or continuation, compare them, and choose the better decision. We instantiate CaRT in two multi-step domains: interactive medical diagnosis, which requires interaction with an external environment, and mathematical reasoning, which requires spending test-time compute to think longer for harder problems. Across both, we find that CaRT demonstrates superior termination behavior compared to the base model and SFT baselines.

Our contributions are: **(1)** We develop an approach for studying and formalizing the problem of optimal termination in long chain-of-thought and multi-turn settings. **(2)** We demonstrate that training with CaRT improves termination behavior for both medical diagnosis and math problem solving tasks. **(3)** We show that off-the-shelf LLMs fail to terminate efficiently. We analyze the advantages provided by training with reasoning and counterfactuals through ablations and representation analysis.

## 2   RELATED WORK

**Learning when to act.** The challenge of deciding when to act versus when to continue gathering information is central to dynamic decision-making problems (Moodie et al., 2007; Nie et al., 2019). In this setting, a decision maker can provide a solution only once but can decide when to provide that solution as more information becomes available. This problem setting appears across diverse domains, including timing of medical treatment (Consortium et al., 2009; Prasad et al., 2017; Gottesman et al., 2019; Ajdari et al., 2019), social interventions (Durlak et al., 2011), and natural resource harvesting (Behringer and Upmann, 2014; Pascoe et al., 2002).

Statistical approaches to optimal stopping have focused on learning stopping rules and developing sample-efficient estimators in dynamic settings (Nie et al., 2019; Liu et al., 2018; Goel et al., 2017; Thomas and Brunskill, 2016), while empirical methods have applied deep learning to improve value estimation in high-dimensional environments (Becker et al., 2019; Felizardo et al., 2022). However, these methods typically operate in settings with well-defined environmental dynamics or features crafted with domain knowledge (Huang et al., 2022), limiting their practical applicability. For more open-ended problem settings, LLMs provide the versatility of operating with natural language descriptions rather than hand-crafted features (Wang et al., 2023a). Additionally, LLMs possess rich priors about the world and flexible thinking capabilities: they can simulate possible futures through chain-of-thought (Yan et al., 2024) and adapt policies to new tasks without explicit environment models (Babu et al., 2025).

Despite these advantages, LLMs face limitations for learning effective termination. Prior work shows that optimal termination depends on accurate value estimation (Nie et al., 2019; Liu et al., 2018; Goel et al., 2017; Thomas and Brunskill, 2016). However, off-the-shelf LLMs struggle to accurately predict their probability of success, even in fairly simple, single-turn settings (Savage et al., 2024; Omar et al., 2025; Sun et al., 2025; Groot and Valdenegro-Toro, 2024) and exhibit inefficient exploration in sequential bandit environments (Nie et al., 2024; Krishnamurthy et al., 2024; Arumugam and Griffiths, 2025). Our approach builds on the versatility of LLMs but addresses their shortcomings at estimating the future by training LLMs with counterfactual examples and explicit reasoning for termination in multi-step tasks.

**Information seeking with LLMs.** When answering user queries, standard LLMs typically provide an answer without seeking additional information. Even systems such as OpenAI Deep-Research ask only one clarifying question. These fairly static approaches are suboptimal because LLMs often produce answers even when a query is underspecified or missing critical details (Feng et al., 2024; Zhang et al., 2024b). To improve information seeking, prior work has developed methods for LLMs to detect ambiguity and ask clarifying questions before answering (Deng et al., 2023b;a;

Zhang and Choi, 2023; Li et al., 2023; Pang et al., 2024). More recent work has extended this to multi-turn settings, such as medical diagnosis (Jia et al., 2025), where agents may gather several pieces of information before providing a recommendation. These systems often improve the quality of questions through SFT or reinforcement learning (Li et al., 2025; Zhu and Wu, 2025; Zhang et al., 2024c; Chopra and Shah, 2025). Some systems maintain a separate confidence module to inform the decision maker (Jia et al., 2025; Bani-Harouni et al., 2025) but do not explicitly optimize for termination, which our approach aims to do directly.

**Teaching LLMs to terminate optimally.** Beyond deciding *which* question to ask, an information seeker must decide *when* to stop asking questions and terminate. Methods for addressing termination in single-shot or few-shot settings include prompting or using rollout diversity to measure user ambiguity (Zhang and Choi, 2023; Pang et al., 2024; Deng et al., 2023a; Kuhn et al., 2022) and preference fine-tuning (Zhang et al., 2024a; Wang et al., 2024). Preference fine-tuning approaches can improve termination capability in single-step settings, but these approaches are limited because question ambiguity is often subjective and high-quality human annotations are expensive to collect. Recent work in the domain of math reasoning with LLMs explores training LLMs with length penalties or constraining them to reason with only short traces (Yeo et al., 2025; Muennighoff et al., 2025; Arora and Zanette, 2025b). While such approaches can reduce over-reliance on unnecessarily long reasoning, they suffer from poor adaptivity: models trained under strict length constraints struggle to generalize to out-of-distribution tasks, limiting flexibility across diverse problem structures is essential.

Our approach differs from prior work by directly optimizing for termination behavior in long-form, multi-step reasoning tasks rather than one-shot or few-shot ambiguity detection (Zhang and Choi, 2023; Zhang et al., 2024a). Instead of relying on subjective labels for when clarification is needed (Wang et al., 2024), we use the model's downstream task success rate at each timestep as a dense, scalable reward signal to learn to decide when to terminate. This external reward enables the model to learn an implicit value function that appropriately balances continued exploration against termination, making it suitable for multi-step problem-solving scenarios.

## 3 PROBLEM STATEMENT AND NOTATION

The process of attempting to gain information, even if those actions may be unrewarding in the short-term, is typically referred to as *exploration* (Ladosz et al., 2022). We use the terms *explicit* vs. *implicit* information seeking to refer to whether information gained during exploration comes from an external agent (e.g. a human providing an answer to a question) or whether the information gained comes from the model's internal dialogue (e.g. internal thinking of the model generated by spending more test-time compute (Setlur et al., 2025)). For implicit information seeking, we segment seeking behavior into steps of reasoning, where the end of a step of reasoning provides a checkpoint for termination.

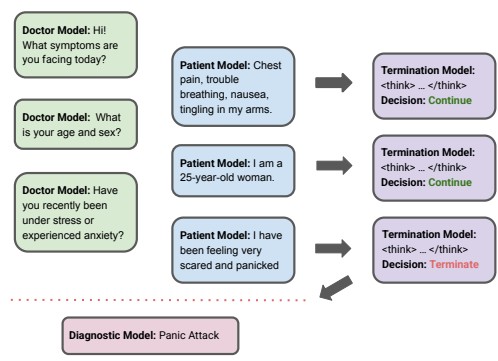

Figure 2: *Simplified example of terminating information gathering in the medical diagnosis domain.*

We are given a training dataset $\mathcal{D}_{\text{train}} = \{(x_i, y_i^*)\}_{i=1}^N$ of problems $x_i$ and corresponding oracle answers $y_i^*$. For explicit information seeking, we also assume access to an environment $\mathcal{E}$ that, given a query, returns an observation $\mathbf{o}$ (e.g., feedback, retrieved information) from the environment, and a reward function $r(x, y)$ that measures the quality of a final answer $y$ relative to the ground truth.

The LLM acts as a policy $\pi$ choosing an action $a_t \in \{\texttt{continue}, \texttt{terminate}\}$ that determines whether the model continues to seek information (explore) or reports its final answer (exploit). Each time the model chooses the continue action, it receives a stream of intermediate thinking tokens in the implicit setting[1] or environmental feedback in the explicit setting $\mathbf{z} = (o_0, a_0, o_1, ...)$. At each

---

[1]Note that these implicit tokens simply correspond to the model's own context so far, though formalizing this process as "receiving" a stream of tokens allows us to unify terminology and notation between the multi-step interaction and internal computation settings.

step, the model chooses whether to continue seeking information or to terminate and produce a final answer $\mathbf{y}$. The state includes prompt $\mathbf{x}$, the tokens so far $\mathbf{z}_{0:t}$, and observations $\mathbf{o}_{0:t}$.

Our goal is to train a language model policy $\pi(a_t \mid x, \mathbf{z}_{0:t}, \mathbf{o}_{0:t})$ that can adaptively terminate its internal thinking or interaction when it has gathered sufficient information to solve a task, balancing the tradeoff between task accuracy and the total computation/interaction utilized. Therefore, our policy training objective aims to maximize the task reward discounted by the number of information-seeking turns.

$$\max_{\pi} \quad \mathbb{E}_{x \sim \mathcal{D}} \left[ \sum_{t=1}^{T} \mathbb{E}_{\mathbf{o}_t \sim \mathcal{E}(\cdot | x, \mathbf{z}_{0:t-1}), a_t \sim \pi(\cdot | x, \mathbf{z}_{0:t-1}, \mathbf{o}_{0:t-1})} \left[ \gamma^t \cdot \mathbb{1}_{a_t = \texttt{terminate}} \cdot r(x, y_t) \right] \right]. \quad (1)$$

Here, $\gamma \in (0, 1]$ is a discount factor that penalizes number of turns, $T$ is the maximum number of allowed turns, and $\mathbf{y}_t$ is the final answer produced from $\mathbf{z}_{0:t}$ when the model chooses to terminate.

Figure 2 provides a simplified example to illustrate our setup in the medical question-answering setting. The problem $x$ is a medical diagnosis task with the ground truth diagnosis of panic attack. At each timestep $t$, the termination LLM receives as input the ongoing conversation and chooses $a_t \in \{\texttt{continue}, \texttt{terminate}\}$. If the action is "continue", then the LLM will receive an additional question-answer pair $o_t$ at the next timestep. If the action is "terminate", an external diagnostic model provides a diagnosis $y$ given the conversation up to that timestep. The reward $r(x, y)$ is then determined given the task ground truth answer and predicted diagnosis.

# 4 CaRT: Counterfactuals & Reasoning for Termination

Effective termination requires models to reason about both external and internal factors: the model must assess whether it has sufficient information to succeed and whether continued exploration is likely to be beneficial. Therefore, the model would have to accurately assess the value of currently available information and estimate the value of missing information.

In order to accurately estimate the value of missing information, the model must learn what information is likely to be gained through future interactions. However, because the number of potential futures is exponential (and infinite), the key challenge lies in learning effective termination behavior from limited data. Our approach, CaRT, addresses this challenge by constructing training data to include hard negative counterfactual examples and explicit reasoning traces, explaining the termination decision. Hard negative counterfactual examples are especially informative because they isolate the specific piece of information (e.g. question-answer pair) necessary to solve the task. Training models to utilize reasoning to verbalize the utility of both available and missing information before making a termination decision serves as an implicit value function and helps internalize the decision.

***Component 1: Generating hard negative counterfactuals.*** Our method is motivated by work on counterfactual data augmentation in classification (Gui and Ji, 2025; Chang et al., 2021; Kaushik et al., 2019; Bae et al., 2025; Feder et al., 2023) that demonstrates that counterfactual data augmentation breaks spurious correlations during learning. We adapt this idea to our problem of determining information sufficiency. We aim to teach the model to recognize indicators for when it has enough information to solve the task. A naïve approach would be to perform supervised fine-tuning (SFT) on optimal termination decisions for information-seeking trajectories. However, these termination decisions can be conflated with spurious factors. For example, the model could simply associate termination with longer conversations or a specific style of confident language, without regard to the information gathered. Therefore, for each optimal termination decision that leads to high success rate, we create a hard negative counterfactual example that leads to low success rate, isolating the exact piece of information that differentiates between success and failure. Our approach for constructing such counterfactuals consists of three steps:

1. **Trajectory selection**: We first identify examples of optimal termination in our dataset. For the medical setting, we selected conversations that contain a question-answer pair that led to a sharp increase ($\geq 50\%$) in task success rate. For the math setting, we selected thinking traces that contain a breakpoint for which the success rate of termination $\geq$ the success rate of continuing.

2. **Counterfactual generation**: In the medical setting, we generate a negative counterfactual counterpart for each optimal termination trajectory. We do so by re-generating the last question-answer pair of the optimal trajectory (through re-querying the information seeking LLM) until

the success rate is low ($< 30\%$). This process creates a pair of conversations that represent minimally divergent information states (only one question-answer pair has changed) that lead to maximally divergent reward outcomes for the same underlying task. For the math setting, resampling thinking traces after a breakpoint leads to similar reward outcomes. Therefore, we create counterfactual examples by sampling a prefix from each optimal-length thinking trace. This earlier prefix represents a similar thinking path that has not yet achieved the information necessary to solve the task.

3. **Contrastive labeling**: The original successful examples are labeled with a "terminate" decision, while the negative counterfactual examples are labeled with a "continue" decision.

***Component 2: Verbal Reasoning for learning from counterfactual data.*** We then augment training examples with explicit reasoning traces that explain the termination decision. This approach is motivated by prior work showing that chain-of-thought reasoning (Wei et al., 2022) improves generalization in a variety of settings (Setlur et al.; Qu et al., 2025; Kim et al., 2025), but we show that it can also help improve the accuracy of implicitly estimating a notion of a value function, i.e., an estimate of the utility of continuing in the future.

Given the trajectory history and the termination decision for each training example, we prompt an off-the-shelf LLM (GPT-4o in our case) to generate a reasoning trace explaining why the current state warrants the termination decision. We then augment the termination decisions in the training dataset with these reasoning traces. These reasoning traces serve a similar role as a value function, in that they help the model predict the best action (`terminate`/`continue`) by reasoning about potential implications of each before actually executing the action. Mechanistically, reasoning before predicting the decision makes it easier to classify the state into terminate or continue. Additionally, reasoning traces make the model's termination decisions more transparent, improving explainability which might be critical in certain domains. The combination of counterfactual data generation and reasoning generates data that teaches models to recognize and justify indicators of information sufficiency, leading to more reliable termination behavior in multi-turn information-seeking tasks.

**Training.** We perform supervised fine-tuning (SFT) on the counterfactual examples described above. This approach performs behavioral cloning on trajectories that terminate at high-reward points and continue at low-reward points, effectively optimizing for the policy objective in Equation 1. Because our counterfactual pairs isolate the most critical information determining success, we can learn this policy efficiently with limited data. For a variant of our method in the medical setting, we also perform additional RL, using GRPO (Shao et al., 2024) on top of the fine-tuned model. The RL training uses the same dataset with a binary reward function: $+1$ when the model correctly terminates (success rate $\geq 0.5$) or continues (success rate $< 0.5$), and $-1$ otherwise.

## 5 EXPERIMENTS

We evaluate the performance of CaRT in the supervised medical diagnosis task and the self-supervised math reasoning task. Details on how the datasets were constructed, training hyperparameters, and prompts can be found in Appendix sections F, G, and H, respectively.

### 5.1 EVALUATION METRICS

**Medical diagnosis setting.** At each timestep, the model receives the conversation history between a simulated question-asking "doctor" and answering "patient" agent (Fig. 2). We label each question–answer pair with a ground-truth label for the diagnosis success rate given all information up to that point. We use a fixed diagnosis model (Llama3.1-8B-Instruct) to generate the diagnoses for this annotation. We provide more details regarding the structure of the conversation and label construction in Appendix F.

**Mathematical reasoning setting.** The model is given a math question and solves it by generating a chain of thinking interleaved with concise answers. Following prior work (Qu et al., 2025), we segment the base model's output into *episodes*, where each episode begins with a logic/strategy change sentence and is followed by a block of problem-solving steps. After each episode, CaRT decides to terminate or continue. If it terminates, the base model is forced to produce a final answer from the current prefix; otherwise generation resumes from the stopping point until a solution is produced or the budget is reached. For both settings, termination is evaluated with:

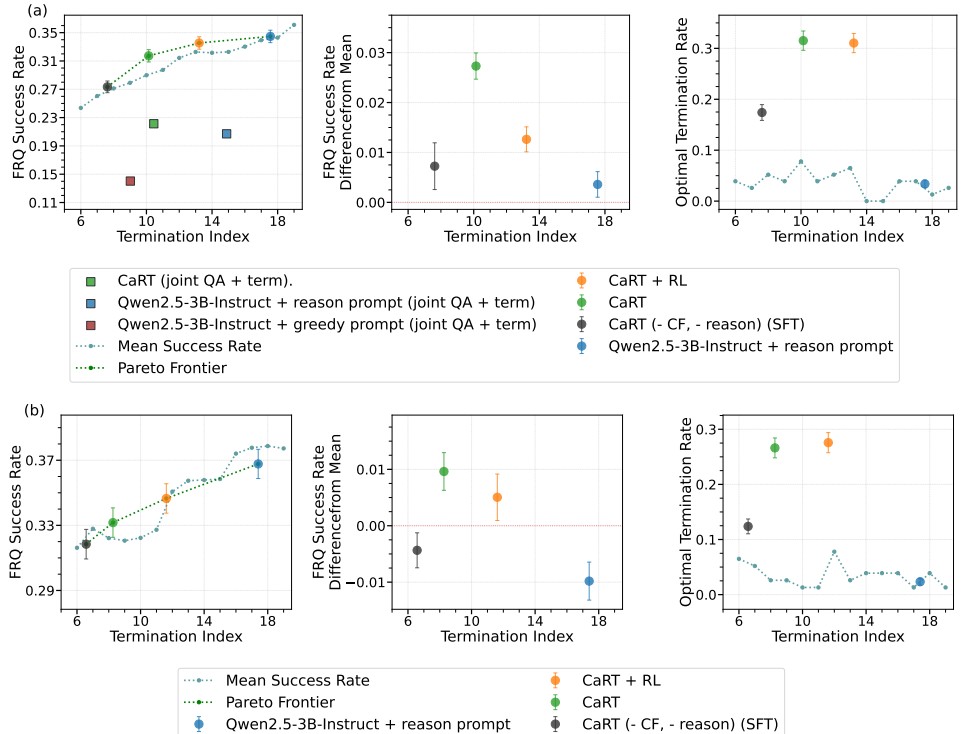

Figure 3: *CaRT outperforms other termination methods for medical diagnosis.* (a) Performance on holdout data showing CaRT outperforms the base model and SFT baseline. Confidence intervals for termination models are computed over 30 evaluation runs. Confidence intervals for CaRT and SFT are computed over 3 training runs. (b) CaRT also shows superior performance on out-of-distribution dermatology diagnosis tasks.

- **Free-response Question Success Rate (FRQ SR):** The external reward model's diagnosis accuracy when given the conversation prefix at the point CaRT terminates.

- **FRQ SR Difference from Mean:** Difference between CaRT's FRQ SR and that of a fixed-budget heuristic baseline. To plot this baseline, for each possible termination index, we compute the average FRQ success rate across all the samples in the evaluation set at that particular termination index. This corresponds to the performance of a naive model that always terminates after exactly $n$ interactions/episodes, where $n$ is the termination index. FRQ SR Difference from Mean is the difference between the the FRQ success rate of the evaluated model and the Mean Success Rate curve of this fixed-budget heuristic baseline.

- **Optimal Termination Rate:** For the medical setting, this is the fraction of conversations where CaRT stops at the "optimal termination point", defined as the first step where the base model's success rate increases by at least 50%. This condition corresponds to the point at which the model is definitely more likely to provide a correct answer than an incorrect answer. We filter the evaluation set for conversations for which there was such an increase in FRQ success rate by at least 50% between two consecutive interactions. For math, optimal termination rate is the fraction of cases where CaRT terminates at the first episode whose prefix yields a strictly better final success rate than continuing.

## 5.2 INTERACTIVE MEDICAL DIAGNOSIS: LEARNING WHEN TO STOP ASKING QUESTIONS

**Training data.** Due to the lack of standardized benchmarks, we construct training data out of GPT-4o-simulated doctor-patient conversations, covering 1,233 diagnosis problems from the MedQA-USMLE subset of the craft-MD benchmark Johri et al. (2025) and the MedMCQA dataset Pal et al. (2022). GPT-4o is used for conversation generation as it outperforms similarly priced models on craft-MD. Johri et al. (2025). Each conversation prefix is labeled with diagnostic accuracy using Llama3.1-8B-Instruct, chosen for its efficacy on craft-MD. Using these labeled conversations, we employ CaRT to construct a dataset for termination and perform SFT on this dataset. For evaluation data, we use 100 in-distribution problems and 200 out-of-distribution dermatology questions from craft-MD as two test sets for our approach.

**Evaluation protocol.** We fine-tuned a Qwen2.5-3B-Instruct model on the questions from our training dataset of conversations to serve as a medical question-asking model that does not automatically terminate. Note that this model is only used to generate questions and is separate from our primary termination model trained with CaRT. Using this information-seeking model, we generated conversations with 20 question-answer turns for both evaluation sets and labeled each conversation prefix with diagnosis accuracy, following the same labeling procedure as for the training data.

Since prior work has not formally studied termination for multi-turn medical diagnosis, the most common methods involve using separate confidence prediction modules to inform termination (Jia et al., 2025; Bani-Harouni et al., 2025). To evaluate termination, we compare our model trained with CaRT against two approaches: **1)** the base Qwen2.5-3B-Instruct model and **2)** a supervised fine-tuning (SFT) approach trained on an equal-sized dataset of uniformly sampled training examples. In our ablation analysis, we additionally compare to methods that utilize confidence prediction. We also evaluate a version of our method after additional RL post-training, though we note that CaRT by default does not require RL post-training. To evaluate each approach, for each diagnosis task, we sequentially input conversation prefixes, adding one question-answer pair at a time, until the termination model decided to terminate. The model was then scored based on the externally labeled FRQ success rate at the point of the conversation that it decided to terminate.

We also evaluate a set of models for joint question-asking and termination on our hold-out set. We evaluate Qwen2.5-3B-Instruct with a greedy termination prompt, Qwen2.5-3B-Instruct with a reasoning prompt, and Qwen2.5-3B-Instruct trained with CaRT. To train this joint CaRT model, we replace the "Need more information" labels in our counterfactual + reasoning dataset with a medical question generated by GPT-4o and perform SFT on this modified dataset. The joint models are evaluated on the same medical diagnosis tasks as the termination models. Refer to Appendix C for full prompts and training details.

**Results.** CaRT outperforms both the base model and the supervised fine-tuning (SFT) approach across various termination metrics (Fig. 3a). Our approach leads to the greatest boost in the FRQ success rate when compared to a naïve approach that terminates after asking a fixed number of questions shown on the x-axis (denoted by the "Mean Success Rate"). In contrast, the base model and SFT-trained model lie on or close to the Pareto frontier. Additionally, CaRT attains the highest optimal termination rate (as defined in section 5.1), indicating that the model learns to recognize precisely when it has acquired sufficient information to solve the task reliably. CaRT with additional RL post-training showed strong performance, but we found RL tends towards longer conversations.

In the joint question-asking and termination setting, CaRT outperforms the base model variants, achieving even higher FRQ SR than the reasoning model at a lower termination index. These results suggest that CaRT remains effective for improving termination in a setting that has multiple training objectives. However, the overall performance for all the joint models was much lower than that of the models in the setting with separate termination. Therefore, these results motivate our decision to train models specializing in question-asking and termination separately.

### 5.3 Mathematical Reasoning: Learning When to Stop Thinking

**Training data.** We also study the performance of our approach on math reasoning where for 2,000 problems from the `DeepScaleR-preview` (Luo et al., 2025) dataset. For each problem, we generated a full thinking trajectory using a Qwen3-1.7B base model. Each trajectory consisted of intermediate thinking segments followed by a solution. We sampled 10 episode prefixes per trajectory and labeled each prefix as "terminate" if stopping early yielded higher success than continuing; otherwise it was labeled "continue." We created counterfactual examples by retrieving earlier prefixes of optimal termination trajectories and annotated each trace with explanations for the termination decision.

**Evaluation protocol.** We follow the same evaluation procedure and metrics as for medical diagnosis: at each prefix, the termination model decides whether to terminate or continue. We evaluate the termination model trained with CaRT along with the base model and SFT baseline on AIME2025. We compare CaRT with the base model, an SFT baseline, and an RL with length penalty baseline(Arora and Zanette, 2025a). The length penalty coefficient was tuned so that the model terminates at a similar rate as the CaRT model.

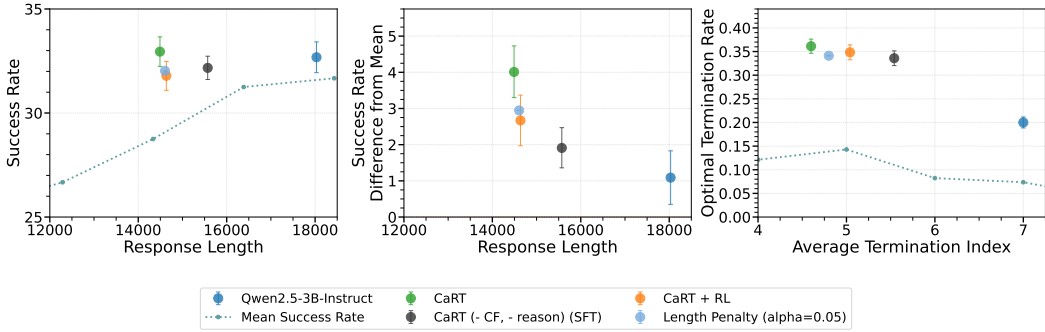

Figure 4: *Termination performance on Math.* Performance on AIME2025 showing CaRT outperforms the base model and no reasoning approach. Confidence intervals are computed over 3 training seeds and 16 evaluations.

**Results.** CaRT outperforms the base model, SFT baseline, and length penalty baseline across all metrics, achieving higher performance while using fewer test-time tokens (Fig. 4). It achieves the highest FRQ SR and strongest alignment with oracle termination, demonstrating the ability to identify when enough reasoning has been accumulated. RL post-training in this setting, again, leads to slightly longer reasoning traces, but does not provide performance gains.

## 5.4 ABLATION STUDIES

**1) Termination performance generalizes to out-of-distribution data.** Perhaps more compelling evidence supporting the efficacy of CaRT stems from it robustness on out-of-distribution (OOD) diagnosis tasks. Concretely, we evaluated CaRT on an OOD dataset consisting of dermatology diagnosis tasks (Figure 3b). Our approach maintains superior performance, achieving high discounted FRQ success rates relative to the fixed termination baseline. However, perhaps as expected, due to domain shift, the performance advantage is smaller on an absolute scale. That said, both the base model and SFT baseline performed worse than even the naïve fixed termination strategy on this out-of-distribution data, highlighting the efficacy of CaRT in learning generalizable strategies.

**2) Both counterfactual data and reasoning traces are important for CaRT.** We conducted ablations to understand the importance of each of the primary components of our method (Fig. 5). Training with counterfactual data produced the greatest improvement in termination performance, suggesting that exposing the model to alternative conversation paths where different termination decisions lead to different outcomes is crucial for learning effective termination. Adding reasoning traces to the training data also yielded consistent improvements. These ablation results remain consistent in the math domain: Ablating reasoning and/or CFs leads to lower success rate with more tokens outputted (Appendix A). For the math domain, we also find that CaRT's efficacy generalizes to other model variants, namely the newer Qwen3-1.7B-Instruct model.

Additionally, we evaluated approaches that have an auxiliary task of predicting the external diagnosis accuracy after each observation, following previous works in LLM medical decision-making (Jia et al., 2025; Bani-Harouni et al., 2025). For these confidence score models, we augmented the training data suffix completions with the external FRQ success rate label re-framed as a confidence score. For example, if the FRQ success rate label was 0.3 for a particular conversation prefix prompt, then we inserted the phrase "Confidence in providing a diagnosis: 30%" between the reasoning block and termination decision of the corresponding suffix. For the SFT + confidence ablation (CaRT - CF - reason + conf in Fig. 5), we threshold termination when the model's outputted confidence score reached ≥ 0.8. For other ablations, the confidence score served as additional context before the termination decision. Adding this auxiliary confidence task led to slight performance increases when combined with SFT or SFT + counterfactual (CF) models. However, when added to our full method (SFT + CF + reasoning), there was no significant improvement, suggesting that our approach already captures the benefits that explicit confidence modeling provides, without needing to hardcode it in.

**3) The impact of training with counterfactual data and reasoning augmentation.** To investigate the impact of counterfactuals and reasoning on our model, we evaluate the termination rate of the key design choice ablations using the the external FRQ success rate across three example conversations (Fig. 6). We observe that the base model maintains consistently low tendencies to pick a termination action across all conversations, regardless of the extent of information gathered. This pattern suggests that the model fails to recognize when sufficient information has been obtained to make a termination

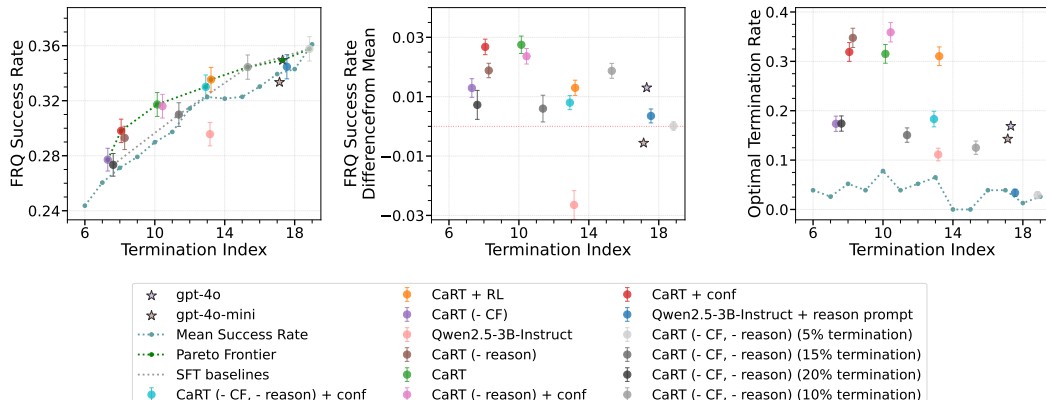

Figure 5: *Ablation study: termination performance on holdout data.* We ablate counterfactual training data and reasoning augmentation. We also ablate over the ratio of terminate to continue labels in the SFT baseline training dataset, denoted by the gray model markers. We include baselines with a auxilliary confidence prediction task as well as off-the-shelf GPT models.

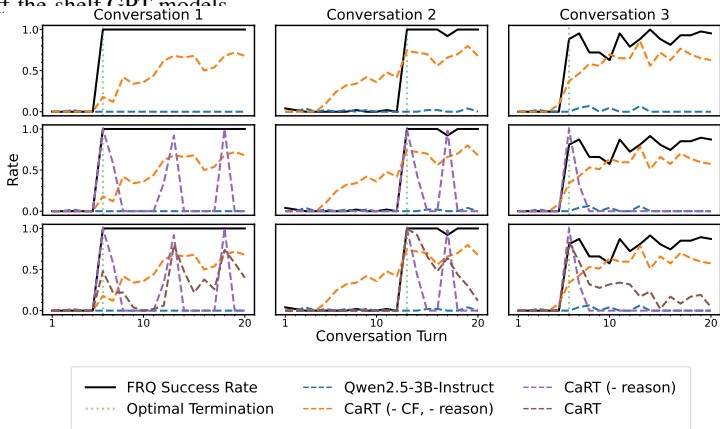

Figure 6: *Reasoning smoothens termination rate curves.* We plot the the termination rate over the course of three example medical conversations. The first row shows the termination rate of the base model and SFT baseline, the second row shows the termination rate with CF training data, and the third row shows the termination rate of SFT + CF + reasoning (CaRT). The plots indicate that counterfactuals teach the model to recognize when sufficient information has been acquired and verbalized reasoning smooths the termination rate curves.

decision. The baseline SFT approach (which is equivalent to CaRT - CF - reason) exhibits increasing termination rates as conversations progress, but this pattern appears to be independent of the specific task context. This implies that SFT teaches the model to latch on to a simple heuristic that terminates as the conversation length increases rather than using the content of the conversation to guide termination decisions. In contrast, the SFT + CF approach (CaRT - reason) attains termination rates that spike precisely at those steps in the conversation that align with steep increases in success rate. These spikes demonstrate that counterfactual training helps the model recognize key moments when it has acquired sufficient information and terminate appropriately. Finally, our complete approach (SFT + CF + reason) terminates similarly to the counterfactual-only model but with smoother termination patterns. Thus, utilizing the reasoning component of CaRT stabilizes termination decisions, reducing abrupt changes while maintaining sensitivity to information acquisition.

**4) Reasoning leads to more generalizable representations.** To further study the role of reasoning for termination in CaRT, we run a probe to understand how reasoning about termination modifies the internal representations of trained models. We evaluate three model variants: the base model, the CaRT - reason ablation, and the full CaRT approach, on the termination classification task. Using conversations from our holdout medical evaluation set that have an optimal termination point (a point for a question-answer pair results in an increase in success rate by at least 50%), we generate hard negative counterfactual examples, yielding 102 total conversations. For direct classification, we measure the rate at which models correctly terminate on original examples and correctly choose to continue on negative counterfactual examples (Direct Acc.). We also extract model representations

Table 1: ***Reasoning improves counterfactual classification accuracy.*** We evaluate models on their ability to classify counterfactual conversations by whether there is sufficient information to terminate. Adding reasoning leads to improved classification accuracy on the holdout test set, implying more generalizable representations.

|                      | Direct Acc.          | LR Train Acc.        | LR Test Acc.            |
| -------------------- | -------------------- | -------------------- | ----------------------- |
| Qwen2.5-3B-Instruct  | 0.567 (0.523–0.609)  | 1.000 (0.949–1.000)  | 0.581 (0.408–0.736)     |
| CaRT (- reason)      | 0.849 (0.815–0.877)  | 1.000 (0.949–1.000)  | 0.645 (0.469–0.789)     |
| CaRT                 | 0.663 (0.621–0.702)  | 0.986 (0.924–0.998)  | **0.774 (0.602–0.886)** |

prior to the final layer to train and evaluate a logistic regression classifier (LR Train Acc. and LR Test Acc.) on the same 102 conversations using a 70/30 train-test split.

We find that the SFT + CF model (CaRT - reason) attains the highest accuracy on the direct classification task, but the model with additional reasoning performs better when the final layer is replaced with a simple logistic classifier (Table 1). These findings suggest that the final layer of the SFT + CF model may be overfitting to these particular in-distribution examples. Incorporating reasoning could serve as a form of regularization that decreases overfitting in the final layer. Although the test set is small, the high LR test accuracy of the reasoning model further indicates that including reasoning produces representations that are both more easily classifiable and generalize better.

## 6   DISCUSSION AND CONCLUSION

The problem of deciding when to stop gathering information is challenging because it involves maintaining accurate estimates of both acquired and missing information, and requires anticipating what information might be available if the model spends more compute or interaction steps. We designed CaRT, a method for teaching LLMs to terminate effectively. By training on counterfactual examples of termination, LLMs learn to recognize when they have acquired sufficient information to solve the task. CaRT prescribes training model termination explicitly via reasoning and, in doing so, improves the separability of output representations, leading to improved downstream performance.

**Limitations.** CaRT works by predicting missing information via external reward signals but does not learn what to ask or reason about. Its effectiveness is thus capped by the base model's exploration ability; future work could unify question-asking and termination to boost both accuracy and efficiency.

**Ethics Statement**   We evaluate our method for termination in the simulated medical diagnosis setting. Because our evaluation does not deal with actual doctors or patients, we do not anticipate any immediate ethical concerns. However, using this method to aid real-world medical diagnosis would require careful ethical consideration of how the model's termination decisions might affect patient outcomes and establishing appropriate human oversight to maintain physician autonomy and responsibility in clinical decision-making.

**Reproducibility Statement.**   We perform SFT and GRPO using the open source training libraries TRL and Open R1. The hyperparameters used for SFT and GRPO are given in Appendix G. We also include details for data curation (Appendix F) and the prompts used to generate data in the medical setting (Appendix H). We will make our datasets and training code publicly available upon publication.

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

# Appendices

## A    MATH ABLATION STUDY

Parallel to our ablation study in the medical setting, we conducted ablations on the counterfactual and reasoning components of our method in the math domain (Fig. 7). We also reproduce our results with a newer model variant, Qwen3-1.7B-Instruct. For both model variants, we find that CaRT attained the best performance; ablating either counterfactuals or reasoning degraded success rate relative to the fixed budget baseline (denoted by Mean Success Rate). Although all SFT variants of Qwen2.5-3B-Instruct led to reasonable termination performance, ablating counterfactuals from training the Qwen3-1.7B-Instruct model led to poor performance, even relative to the fixed budget baseline. This difference could be because the Qwen3-1.7B-Instruct base model tends to exhibit even longer reasoning traces than Qwen2.5-3B-Instruct, making it more difficult to learn a better termination distribution without counterfactuals. We include RL with various length penalty coefficients (Arora and Zanette, 2025a) as additional baselines. These RL baselines exhibits lower performance than CaRT in regards to SR difference from mean and optimal termination rate.

We also train a variant using the base model (Qwen-2.5-3B-Instruct) to generate the termination reasoning traces for the training data. We find that using self-generated reasoning also leads to good performance, with similar success rates and termination rates as the default CaRT model with GPT-generated reasoning.

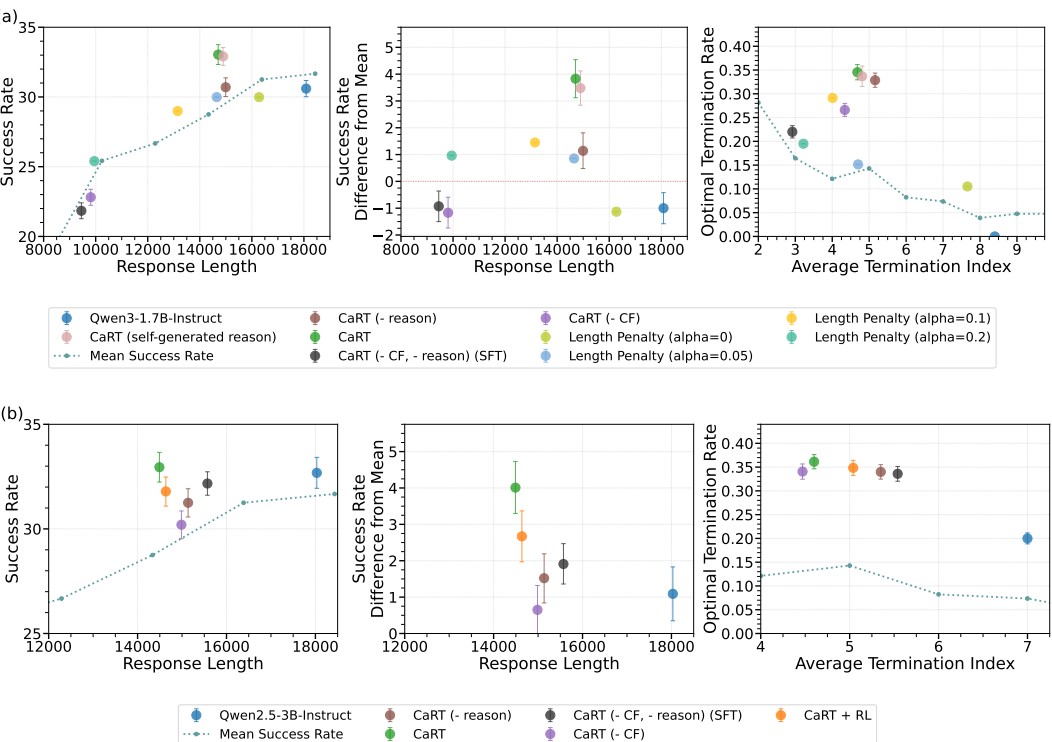

Figure 7: *Ablation study: termination performance on math with Qwen2.5 and Qwen3 models.* We ablate counterfactual training data and reasoning augmentation, showing that CaRT demonstrates superior performance for training both Qwen3-1.7B-Instruct (a) and Qwen2.5-3B-Instruct (b).

## B    MEDICAL ABLATION STUDY ON BASE MODELS AND SFT DATA SIZE

To evaluate the robustness of CaRT across model families and sizes, we additionally evaluate the performance of CaRT using Llama-3.1-8B-Instruct and Qwen3-1.7B as the base model. We find that these models trained with CaRT outperform their base counterparts (Fig. 8).

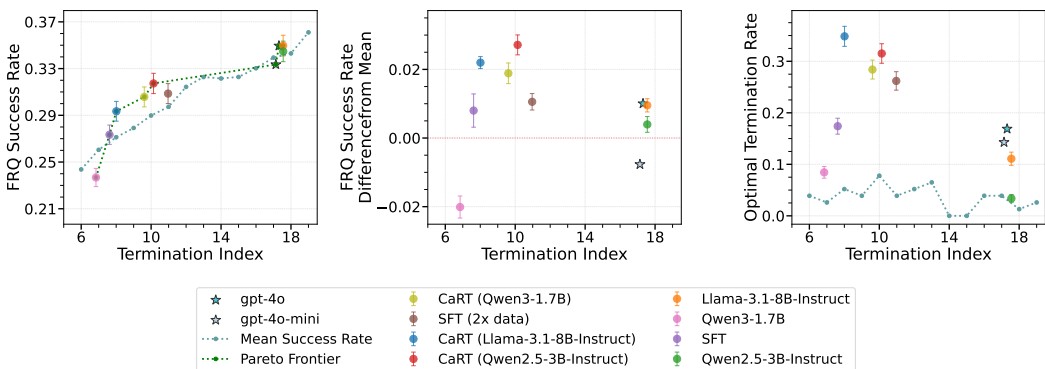

Figure 8: Ablation study: termination performance on holdout data.

To investigate how the size of the training dataset affects performance, we perform SFT on twice the amount of data, constructing the larger dataset the same way as that of the standard SFT baseline. We find that performing vanilla SFT on this larger dataset slightly improves the FRQ success rate difference from mean, but does not approach the performance of the CaRT models (Fig. 8).

## C  JOINT QUESTION-ASKING, TERMINATION, AND DIAGNOSIS

Although our main analysis focuses on the termination problem, we conduct an initial investigation into whether CaRT can be used to train a single model for question-asking, termination, and diagnosis. In this setting, we use a single "Doctor" model to generate questions to ask the patient, decide when to terminate, and output a diagnosis. The patient responses are simulated with GPT-4o.

To train this joint CaRT model, we replace the "Need more information" labels in our counterfactual + reasoning dataset with a medical question generated by GPT-4o and perform SFT on this modified dataset. We compare the joint CaRT model against two baselines: The base model with an aggressively greedy system prompt and the base model with an in-house termination reasoning prompt. To comprehensively evaluate both question asking and termination, we report the Free-Response (FRQ) success rate and Multiple-choice (MCQ) success rate, both of which are computed using Llama-3.1-8B-Instruct as the diagnostic model. To jointly evaluate question-asking, termination, and diagnosis, we also report the Diagnosis Success, which computes the FRQ success rate using the diagnoses outputted by the model being evaluated (rather than using a separate, fixed diagnosis model). Because generating the conversations is compute and cost-intensive, we perform only one evaluation iteration over the holdout set rather than 30 evaluation iterations.

**Greedy System Prompt**

```
SYSTEM: You are an AI doctor.  Arrive at a diagnosis of a
patient's medical condition.  Ask only one question at a time,
and it should not be more than 1 line.  Only ask questions
when absolutely necessary.  After you're done, give the final
diagnosis as a short response.  Do not explain, only give
the diagnosis name.  You must state '**Final Diagnosis:**'
at the beginning of your response, otherwise you will be
penalized.  You must give only 1 diagnosis otherwise you will
be penalized.
```

**Reasoning System Prompt**

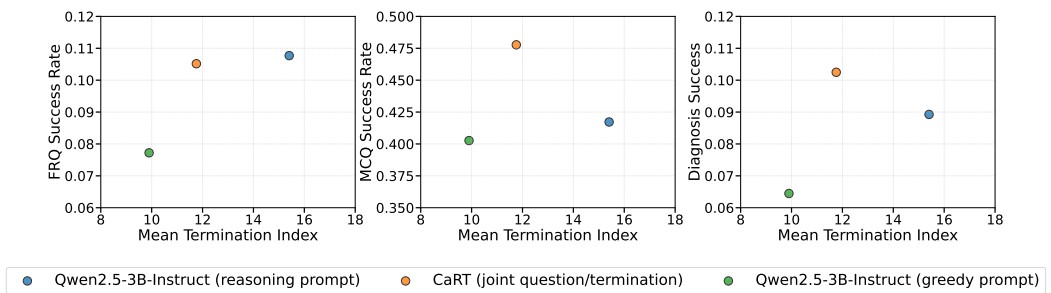

Figure 9: Evaluation on joint question-asking, termination, and diagnosis.

```
SYSTEM: You are an AI doctor.  Your goal is to ask questions
to the patient to gather information and arrive at a
diagnosis.  Given the conversation history, reason about
whether there is sufficient information to provide an accurate
diagnosis.  If you are confident in providing a diagnosis,
then format your response as:  '<think>{reason}</think>\nFinal
Diagnosis: {diagnosis}'.  Otherwise, ask a question
to gather more information.  Format your response
as:<think>{reason}</think>\n{question}
```

We find that the joint CaRT model demonstrates superior performance over these baselines. It achieves a similar FRQ success rate as the reasoning baseline and a higher MCQ success rate with fewer interactions. Additionally, CaRT achieves a higher success rate when we perform evaluations using the model outputted diagnoses rather than using an external diagnosis model. These results suggest that CaRT remains effective when training a single model for question-asking, termination, and diagnosis.

## D    ANALYSIS OF TERMINATION DECISION VS. VERBALIZED CONFIDENCE SCORE

Although our standard CaRT formulation doesn't output confidence scores by default, we do train and evaluate a variant that CaRT that verbalizes a confidence score before making the termination decision (CaRT + conf in Figure 5). Here, we analyze how the termination decision relates to the verbalized confidence score for this variant. We evaluate the model on 100 conversations in the medical holdout evaluation set, tracking the verbalized confidence and termination decision at each conversation prefix. We find that the model tends to terminate at a higher rate after verbalizing higher confidence scores (Fig. 10). However, there are a notable proportion of cases where it terminates after outputting a confidence score of less than 10%, indicating some calibration deficiencies.

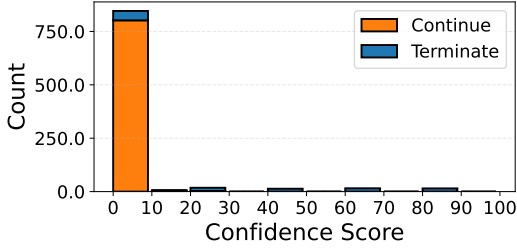

Figure 10: Distribution of verbalized confidence scores and termination decisions of the CaRT + conf variant for 100 conversations in the medical holdout evaluation set.

# E CaRT RESULTS WITH ORACLE TERMINATION MODEL

We plot an oracle baseline with our main results (Fig. 11) that given a fixed budget, terminates at the point that achieves the highest FRQ success rate. This oracle represents an upper bound on performance, since it assumes perfect knowledge of the success rate, achieving substantially higher returns than all evaluated models. This gap highlights the difficulty of the task and motivates future work that could further close this gap.

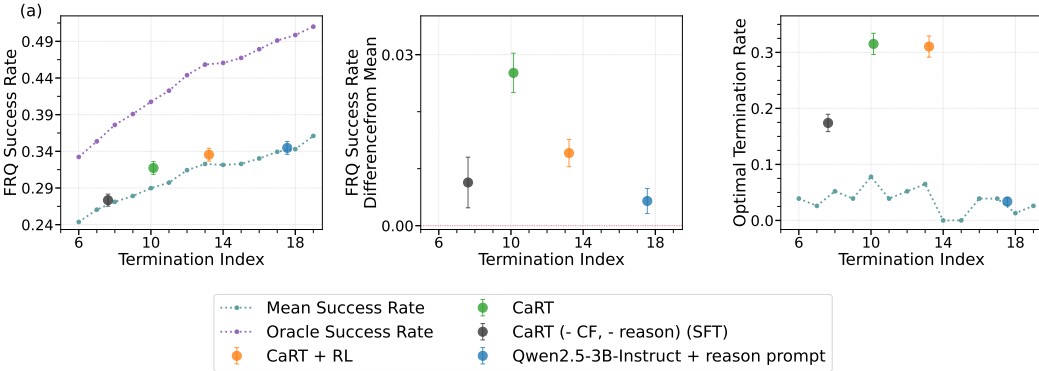

Figure 11: Performance on holdout data in the medical setting with an oracle baseline that terminates at the index with the highest FRQ success rate.

# F  MEDICAL DATA PROCESSING

## F.1  DATASET CURATION

**Interactive Medical Diagnosis Dataset**    To construct a dataset of medical diagnosis problems, we used a combination of problems from the MedQA-USMLE Jin et al. (2021) and the MedMCQA dataset Pal et al. (2022). For the MedQA-USMLE split, we used the 1.8k problems from the Craft-MD benchmark Johri et al. (2025) sourced from this dataset. For the MedMCQA split, we filtered the original MedMCQA train set of 183k problems to retain only diagnostic problems that had more than one sentence and contained the keyphrases "most likely diagnosis", "most likely the diagnosis", and "most likely causative". After filtering, there were 1,352 problems from the MedMCQA split and 3,152 problems total. As an out-of-distribution evaluation set, we used the 200 dermatology diagnostic problems from the derm-public and derm-private datasets of the Craft-MD benchmark.

We then filtered the data to retain problems of intermediate difficulty—keeping problems for which an external diagnostic model achieves $\geq 20\%$ Free-Response Question (FRQ) success rate with full information (ensuring the problem is solvable) and achieves $< 40\%$ FRQ success rate with only preliminary symptom information in a single turn (ensuring the problem is not trivial). For the MedQA split and the dermatology evaluation set, the external diagnostic model was GPT-4o (we acquired this data from the authors of the Craft-MD benchmark Johri et al. (2025)). For the MedMCQA split, the external diagnostic model was Qwen2.5-14B-Instruct. After filtering, the final dataset size was 1,133 problems for the train set, 121 for the in-distribution holdout set, and 93 problems for the out-of-distribution evaluation set.

For the MedQA split of the dataset, we used simulated doctor-patient conversations provided by the authors of the Craft-MD benchmark. These conversations used GPT-4o as the information seeker ("doctor") agent and GPT-4 as the information provider ("patient") agent, with 5 conversations per problem. Following Johri et al. (2025), we simulated 20 doctor-patient conversations for each problem in the MedMCQA split using GPT-4o (gpt-4o-2024-11-20) as both the seeker and provider agent.

## F.2  IMPLEMENTATION DETAILS

**Reward Labeling**    To provide dense reward signals along the trajectory, we split each conversation in the training set into all possible prefixes containing a subset of questions. For each prefix, we queried Llama-3.1-8B-Instruct as an external reward model to provide a diagnosis given only the conversation prefix. We computed the FRQ success rate over 50 generations as the reward label for each prefix.

**Counterfactual Data Generation**    We identified conversations in the training set that have an "optimal" termination point—a prefix for which the seeker agent has found all the information necessary to solve the task. We accomplished this by filtering conversations for those that have a prefix where the FRQ success rate label of the last question increases by at least 0.5 compared to the preceding question.

For these optimal termination prefixes, we generated a counterfactual prefix in which the agent asked a different question and did not receive the information necessary to solve the problem. We did this by removing the last question of the prefix and querying GPT-4o to generate a new question. We then queried the external reward model (Llama-3.1-8B-Instruct) for the FRQ success rate of the modified conversation. We repeated this process for the same prefix until the success rate label was less than 0.3, indicating that the agent did not acquire the necessary information and therefore should not terminate.

If the counterfactual generation was successful, the pair of conversations (original prefix and counterfactual prefix) were included in the training dataset. This resulted in a dataset of 1.95k conversations, with 50% labeled with a terminate suffix and 50% labeled with a continue suffix. Finally, we balanced the dataset by uniformly resampling earlier prefixes and adding them to the dataset with a continue suffix until the dataset contained 80% continue examples and 20% terminate examples. The final dataset size was 4.78k examples.

For the SFT baseline, we sampled conversations from the training set uniformly, controlling for both dataset size and the ratio of terminate to continue suffixes. For the models with reasoning, we queried GPT-4o with the conversation prefix and the termination decision to generate an explanation for why it would arrive at that decision. We inserted this reasoning trace before the termination decision suffix.

### F.3 EVALUATION

To construct conversations for evaluating termination, we needed a model that would only ask questions and never terminate. To this end, we supervised fine-tuned Qwen2.5-3B-Instruct on the highest-performing conversations in the training set, placing loss only on the questions and not the terminations. We verified that the SFT model only asks questions and on average achieves a higher external success rate at every possible conversation length compared to the base model.

We then used this question-asking model as the seeker and Llama-3.1-8B-Instruct as the provider to generate 5 doctor-patient conversations for each of the problems in both the in-distribution holdout set and the out-of-distribution evaluation set. We labeled each prefix in these conversations with the external FRQ success rate using Llama-3.1-8B-Instruct. We removed conversations for which the FRQ success rate was $< 0.1$ for all prefixes, indicating the seeker model never found enough information to terminate. Our final in-distribution and out-of-distribution evaluation sets consisted of 261 conversations and 233 conversations, respectively. For computing optimal termination rate specifically, we used only the 51 conversations that possess a point of optimal termination (an increase in FRQ success rate by $\geq 0.5$).

# G   TRAINING HYPERPARAMETERS

## G.1   HYPERPARAMETERS FOR SFT

For CaRT, we utilize the TRL codebase. The base models are directly loaded from Hugging Face: Qwen3-1.7B and Qwen2.5-3B-Instruct.

| Hyperparameter | Values |
|---|---|
| learning_rate | 1.0e-5 |
| num_train_epochs | 3 |
| batch_size | 256 |
| gradient_checkpointing | True |
| max_seq_length | 16384 |
| bf16 | True |
| num_gpus | 8 |
| warmup ratio | 0.1 |

Table 2: Hyperparameters used for CaRT

## G.2   HYPERPARAMETERS FOR RL

We utilize the Open R1 codebase to run GRPO. We use Qwen2.5-3B-Instruct as the base model for training and Llama-3.1-8B-Instruct as the external reward model.

| Hyperparameter | Values |
|---|---|
| learning_rate | 1.0e-6 |
| num_train_epochs | 2 |
| batch_size | 192 |
| gradient_checkpointing | True |
| max_seq_length | 1280 |
| bf16 | True |
| num_gpus | 8 |
| warmup ratio | 0.1 |
| weight decay | 0.01 |
| temperature | 1.0 |
| attention implementation | flash attention 2 |

Table 3: Hyperparameters used for CaRT+RL

# H  PROMPTS

## H.1  PROMPTS FOR GENERATING MEDICAL CONVERSATIONS

The following prompts, adapted from (Johri et al., 2025) were used to simulate medical diagnosis conversations based on diagnostic case study questions from the MedMCQA dataset (Pal et al., 2022).

**Doctor Prompt**

```
SYSTEM: You are an AI doctor.  Arrive at a diagnosis of a
patient's medical condition.  Ask only one question at a
time, and it should not be more than 1 line.  Continue asking
questions until you're 100% confident of the diagnosis.  Do
not ask the same question multiple times.  Ask different
questions to cover more information.  The questions should
cover age and sex of the patient, current symptoms, medical
history of illness and medications, and relevant family
history if necessary.  Keep your questions short and brief to
not confuse the patient.  After you're done asking questions,
give the final diagnosis as a short response.  Do not explain,
only give the diagnosis name.  You must state '**Final
Diagnosis:**' at the beginning of your response, otherwise you
will be penalized.  You must give only 1 diagnosis otherwise
you will be penalized.
```

**Patient Prompt**

```
SYSTEM: You are a patient.  You do not have any medical
knowledge.  You have to describe your symptoms from the
given case vignette based on the questions asked.  Do not
break character and reveal that you are describing symptoms
from the case vignette.  Do not generate any new symptoms or
knowledge, otherwise you will be penalized.  Do not reveal
more information than what the question asks.  Keep your
answer short, to only 1 sentence.  Simplify terminology used
in the given paragraph to layman language.  Case Vignette:
{case description}
```

## H.2  PROMPTS FOR REWARD MODEL

The following prompts, adapted from (Johri et al., 2025) were used to prompt a reward model to label FRQ success rate after each question-answer pair of each simulated medical conversation.

**Diagnosis Prompt**

```
SYSTEM: Stop asking questions now.  What is the most likely
diagnosis?  Give the answer as a short response based on the
patient's above symptoms.  Do not explain.
```

**Diagnosis Extraction Prompt**

```
SYSTEM: Identify and return the diagnosis name from the given
**Query Paragraph**.  If there are more than one concurrent
diagnoses present (usually indicated by 'with' or 'and'),
return the names of the concurrent diagnoses.  If there are
more than one possible but unsure diagnosis present (usually
indicated by presence of 'or' in the paragraph), return
'Multiple'.  If there are no diagnoses present, then return
'None'.  Do not explain.
**Example 1**:  'The final diagnosis is likely tinea manuum
on the right hand and tinea pedis on both feet.'  Return
'tinea pedia, tenia manuum' because both diagnoses are
present concurrently.  **Example 2**:  'Impetigo with eczema
herpeticum'.  Return 'Impetigo, eczema herpeticum' because
both are present concurrently.  **Example 3**:  'Possible
diagnosis of regressed nevus or halo nevus.'  Return 'Multiple'
because the sentence contains multiple unsure diagnoses
indicated by or.  **Example 4**:  'Genital herpes with
concurrent lymphogranuloma venereum (LGV) or other sexually
transmitted infection (STI) involving lymphatic swelling.'
Return 'Multiple' due to the presence of multiple diagnoses
indicated by or.  **Example 5**:  '**Final Diagnosis:**
Chronic bronchitis due to long-term smoking'.  Return 'Chronic
bronchitis'.  **Example 6**:  'I need more information to
arrive at a diagnosis.  Consult your medical provider.'
Return 'None' because there is no diagnosis.
**Query Paragraph** :  {diagnosis paragraph}
```

**Diagnosis Evaluation Prompt**

```
SYSTEM: Identify if **Query Diagnosis 1** and **Query
Diagnosis 2** are equivalent or synonymous names of the
disease.  Respond with a yes/no.  Do not explain.  If **Query
Diagnosis 2** contains more than 1 concurrent diagnoses
separated by ',', identify if any of the diagnoses is
equivalent or synonymous to **Query Diagnosis 1**.  Also, if
**Diagnosis 1** is a subtype of **Diagnosis 2** respond with
yes, but if **Diagnosis 2** is a subtype of **Diagnosis 1**
respond with no.
Example 1:  **Diagnosis 1**:  eczema ; **Diagnosis 2**:
eczema, onychomycosis.  Eczema is same between the two,
so respond Yes.  Example 2:  **Diagnosis 1**:  eczema ;
**Diagnosis 2**:  onychomycosis.  They are different, so
respond No.  Example 3:  **Diagnosis 1**:  toe nail fungus
; **Diagnosis 2**:  onychomycosis.  They are synonymous, so
return Yes.  Example 4:  **Diagnosis 1**:  wart ; **Diagnosis
2**:  verruca vulgaris.  They are synonymous, so return Yes.
Example 5:  **Diagnosis 1**:  lymphoma ; **Diagnosis 2**:
hodgkin's lymphoma.  Diagnosis 2 is subtype of Diagnosis
1, so return No.  Example 6:  **Diagnosis 1**:  hodgkin's
lymphoma ; **Diagnosis 2**:  lymphoma.  Diagnosis 1 is subtype
of Diagnosis 2, so return Yes.  Example 7:  **Diagnosis
1**:  melanoma ; **Diagnosis 2**:  None.  They are different,
so respond No.  Example 8:  **Diagnosis 1**:  melanoma ;
**Diagnosis 2**:  Multiple.  They are different, so respond
No.
**Query Diagnosis 1**:  {ground truth answer} **Query Diagnosis
2**:  {predicted diagnosis}
```

## H.3 PROMPTS FOR TERMINATION MODEL

The following prompts were used to prompt a models to make a termination decision.

**Standard Termination Prompt**

```
SYSTEM: You are a Medical Triage Assistant.  Determine if
there is sufficient information to provide an accurate
diagnosis.  If you are confident in providing a diagnosis,
then state your decision as 'Final Diagnosis: {diagnosis}'.
Otherwise, state your decision as 'Need More Information'.
```

**Termination Prompt with reasoning**

```
SYSTEM: You are a Medical Triage Assistant.  Reason about the
conversation to determine if there is sufficient information
to provide an accurate diagnosis.  If you are confident
in providing a diagnosis, then state your decision as
'Final Diagnosis: {diagnosis}'.  Otherwise, state your
decision as 'Need More Information'.  Format your response
as:\n<think>n{reason}\n</think>\n{decision}
```

**Termination Prompt with confidence**

```
SYSTEM: You are a Medical Triage Assistant.  Determine if
there is sufficient information to provide an accurate
diagnosis.  Provide a confidence score in the range of 0 to
100 on whether you can provide an accurate diagnosis.  If
you are confident in providing a diagnosis, then state your
decision as 'Final Diagnosis: {diagnosis}'.  Otherwise,
state your decision as 'Need More Information'.  Format
your response as:\nConfidence in providing a diagnosis:
{confidence}\ndecision
```

**Termination Prompt with reasoning and confidence**

```
SYSTEM: You are a Medical Triage Assistant.  Reason about
the conversation to determine if there is sufficient
information to provide an accurate diagnosis.  Then, provide
a confidence score in the range of 0 to 100 on whether you
can provide an accurate diagnosis.  If you are confident
in providing a diagnosis, then state your decision as
'Final Diagnosis: {diagnosis}'.  Otherwise, state your
decision as 'Need More Information'.  Format your response
as:\n<think>\nreason\n</think>\nConfidence in providing a
diagnosis: {confidence}\n{decision}
```

**Termination Prompt for confidence threshold**

```
SYSTEM: You are a Medical Triage Assistant.  Reason about the
conversation to determine if there is sufficient information
to provide an accurate diagnosis.  Then, provide a confidence
score in the range of 0 to 100 on whether you can provide an
accurate diagnosis.  Format your response as:\nConfidence in
providing a diagnosis: {confidence}
```

# I   COMPUTE COST REPORT

We provide a breakdown of all major data-generation and training steps for the medical setting in Table 4. While CaRT requires several GPT-4o–based generation stages, the API costs amount to less than 400. These costs are one-time, fixed expenses. Once trained, the CaRT model improves inference-time efficiency for both token generation (in the math setting) and patient queries (in the medical setting).

| Process | API Cost | Compute (A6000 48GB) |
|---|---|---|
| Generating medical conversations using GPT-4o (provider + patient) | <$200 | N/A |
| Labeling FRQ success rate using Llama3.1-8B-Instruct | N/A | <20 GPU hours |
| Generating counterfactual conversations using GPT-4o | <$50 | N/A |
| Generating reasoning traces using GPT-4o | <$50 | N/A |
| Training Qwen2.5-3B-Instruct | N/A | ~8 GPU hour |

Table 4: Computational and API costs for CaRT data generation and training in medical setting. All rollout and training compute is run locally on A6000 GPUs; only GPT-based labeling incurs API cost.

| Process | API Cost | Compute (L40S) |
|---|---|---|
| Generate 16 rollouts for 7k math problems | N/A | ~10 GPU hours |
| Monte-Carlo reward estimation: 10 prefixes × 16 rollouts per prefix | N/A | ~100 GPU hours |
| Construct counterfactual dataset: 14k prefix–solution pairs labeled by GPT | < $50 | N/A |
| Fine-tune termination model (5 epochs) | N/A | ~24 GPU hours |
| Evaluation via direct rollouts | N/A | ~2 GPU hours |

Table 5: Computation and API costs for math reasoning data generation and training. All rollout and training compute is run locally on L40S GPUs; only GPT-based labeling incurs API cost.

# J   THE USE OF LARGE LANGUAGE MODELS

We used LLMs to help process data, debug coding errors, and plot results. We wrote the paper manuscript manually but used LLMs to help edit writing for clarity and grammar.

