# OpenReview forum: "Teaching LLMs When to Stop Seeking and Start Acting"
_ICLR.cc/2026/Conference — Submitted to ICLR 2026_

### Official Review · Reviewer_VcE9 · 2025-10-18

**Soundness:** 3
**Presentation:** 3
**Contribution:** 3
**Rating:** 6
**Confidence:** 4

**Summary:**

This paper builds an approach to imbue LLMs with the ability to stop or terminate their ternal thinking process and/or environment interaction at the right point for maximal performance, without wasting computation or interaction. The high-level insight of the proposed framework is that reasoning itself can be used to learn accurate and generalizable termination behavior, as long as this reasoning is done comparatively. Then experimental results demonstrate the good performance of the proposed model.

**Strengths:**

1. The paper addresses a critical problem for deploying LLMs: deciding when to terminate. The idea itself is interesting. The proposed idea of combining counterfactuals with reasoning is novel and effective.
2. The related work section is comprehensive, systematically surveying the field from the perspectives of learning when to act, LLM-based information seeking, and optimal termination.
3. The experimental design is comprehensive and convincing. The evaluation across two distinct domains—interactive medical diagnosis and mathematical reasoning—effectively demonstrates both the generality and superiority of the proposed CaRT method.

**Weaknesses:**

1. The training and evaluation pipeline relies heavily on simulated environments and external reward models, which introduces a potential sim-to-real gap.
2. The method focuses solely on the termination policy ("when to stop") and does not address the exploration policy ("what information to seek"), which is a crucial limitation.
3. The training of CaRT itself appears computationally expensive, as it utilizes LLMs to generate counterfactual data and reasoning traces. The paper does not discuss this upfront cost in relation to the achieved inference-time efficiency gains, which is a practical concern for resource-constrained settings.

**Questions:**

1. Figure 1 is not cited in the main text.
2. A key limitation is the separation of termination ("when to stop") from exploration ("what to seek"); unifying them is a critical next step. Can the author discuss how to deal with this problem?
3. Using GPT-4o to generate reasoning traces potentially risks knowledge leakage. The performance gains of CaRT are confounded; it is unclear if they stem from the method itself or from the injected knowledge of a stronger teacher model. An ablation study is needed to isolate the true contribution of the reasoning framework.
4. The paper does not discuss the substantial computational overhead of training CaRT, which involves generating counterfactual data and reasoning traces with GPT-4o. Could the author provide an analysis of this upfront cost versus the achieved inference-time efficiency?
5. The definition of the "optimal termination point" (a ≥ 50% success rate increase) appears arbitrary. This threshold may not exist in cases of gradual information accumulation, causing the "Optimal Termination Rate" metric to be computed on a biased subset of examples. Is it possible to provide a threshold-agnostic metric that evaluates performance across all samples.
6.	Current baselines seem insufficient for a comprehensive evaluation. Adding an oracle baseline that selects actions to maximize future discounted reward at each step would better demonstrate the gap to the theoretical performance upper bound.

---

> ### Author Response · Authors · 2025-11-21
>
> Dear Reviewer,
>
> Thank you for the thoughtful and detailed feedback. It seems like the main concerns relate to practicality of the method in deployment. To address these concerns, we have added discussion and new experiments expanding on the practicality of our method. These updates are included in the revised version of the paper, and we summarize the relevant details below. **If our revisions address your concerns, we would be sincerely grateful if you could consider raising your score.**
>
>
> ---
>
> > The training and evaluation pipeline relies heavily on simulated environments and external reward models, which introduces a potential sim-to-real gap… unifying them is a critical next step. Can the author discuss how to deal with this problem?
>
> We agree that improving termination for more realistic environments is a critical next research step. One avenue to improve the practical value of our method is to demonstrate that CaRT can be used to train a single model for question asking, termination, and diagnosis, reducing the need for an external reward model or external QA model. We perform an initial investigation into this (see comment directly below).
>
> To employ the method in more realistic settings, creating termination training data from real medical conversations is an exciting avenue for future research. Additionally, expanding CaRT to other domains that require effective termination, such as medical intervention or purchasing decisions, could further demonstrate the generalizability of the method.
>
> ---
>
> > The method focuses solely on the termination policy ("when to stop") and does not address the exploration policy ("what information to seek"), which is a crucial limitation.
>
> In our main analysis, we keep the question-asking model separate to focus our evaluation specifically on the termination problem. This choice allows us to avoid the situation in which we observe improved performance of the trained model, but we are unable to distinguish whether the improvement is because the model improved its ability to determine information sufficiency or because the model improved the quality of its questions due to vanilla SFT.
>
> However, we agree that training a joint model for termination and question-asking is an important direction for future work. To address this concern, we have performed a new experiment where we train a single "doctor" model for question-asking, termination, and diagnosis. We find that CaRT demonstrates promising results in this joint setting, achieving similar FRQ success rate and higher Multiple Choice success rate while being more efficient than a baseline reasoning model (Figure 9 in **Appendix C**). Additionally, we evaluate diagnosis accuracy - evaluating the model's outputted diagnoses rather than using an external diagnosis model - and find that the CaRT model demonstrates better diagnosis accuracy (+2.2%) than the reasoning base model, again while asking fewer questions.
>
> That said, we note that prior work has addressed the question of "what information to seek" (Li et al. 2025, Zhu et al. 2025) but has not jointly optimized for termination.
>
> ---
>
> > Could the author provide an analysis of this upfront cost versus the achieved inference-time efficiency?
>
> We have added a cost analysis for generating the training data for both the math and medical setting in **Appendix I**.
>
> ---
>
> > Using GPT-4o to generate reasoning traces potentially risks knowledge leakage... An ablation study is needed to isolate the true contribution of the reasoning framework.
>
> We have added a new experiment to address this concern. We train a variant of CaRT using the base model (Qwen-2.5-3B-Instruct) to generate the termination reasoning traces for the training data in the math setting. We find that using self-generated reasoning maintains good performance, with similar success rates and termination rates as the default CaRT model with GPT-generated reasoning (Figure 7 in **Appendix A**). These results suggest that the efficacy of CaRT does not depend on GPT-4o for generating effective reasoning traces.

---

> > ### Author Response · Authors · 2025-11-21
> >
> > ---
> >
> > > The definition of the "optimal termination point" (a ≥ 50% success rate increase) appears arbitrary... Is it possible to provide a threshold-agnostic metric that evaluates performance across all samples.
> >
> > We agree that the optimal termination point is threshold dependent and limits the number of samples that we evaluate on. We believe that the optimal termination rate still provides value because it shows the the rate at which the model terminates exactly at the point when it acquires very useful information.
> >
> > For a threshold-agnostic metric, we present the downstream FRQ success rate and plot the different between the FRQ success rate of each evaluated model from a heuristic fixed-termination baseline. This metric is computed over all samples in the evaluation set.
> >
> > ---
> >
> > > Adding an oracle baseline that selects actions to maximize future discounted reward at each step would better demonstrate the gap to the theoretical performance upper bound.
> >
> >
> > We have added a new evaluation in **Appendix E** of our method in relation to an oracle baseline that, given a fixed budget, terminates at the point that achieves the highest FRQ success rate. This oracle represents an upper bound on performance, since it assumes perfect knowledge of the success rate, achieving substantially higher returns than all evaluated models. This gap highlights the difficulty of the task and motivates future work that could further close this gap.
> >
> >
> > > Figure 1 is not cited in the main text.
> >
> > Thanks for the pointer, we have cited Figure 1 in the introduction.
> >
> > ---
> >
> > ## References
> >
> > Li, S. S., Mun, J., Brahman, F., Ilgen, J. S., Tsvetkov, Y., & Sap, M. (2025). Aligning llms to ask good questions a case study in clinical reasoning. arXiv preprint arXiv:2502.14860.
> >
> > Zhu, J., Pan, J., Liu, Y., Liu, F., & Wu, J. (2025, November). Ask patients with patience: Enabling llms for human-centric medical dialogue with grounded reasoning. In Proceedings of the 2025 Conference on Empirical Methods in Natural Language Processing (pp. 2846-2857).

---

> > > ### Author Response · Authors · 2025-11-26
> > >
> > > Dear Reviewer,
> > >
> > > We have incorporated the review feedback by running new experiments and revising the paper. We'd really appreciate if you could confirm whether these changes address the concerns about the paper. If we have misunderstood any of the concerns, we'd like to learn so that we can further revise the paper or run additional experiments.
> > >
> > > Best,
> > > The Authors

---

> > > > ### Comment · Reviewer_VcE9 · 2025-11-27
> > > > **Response from reviewers**
> > > >
> > > > I have read the authors' response, which adequately addresses my concerns. I decide to keep my score

---

### Official Review · Reviewer_bJmE · 2025-10-28

**Soundness:** 2
**Presentation:** 3
**Contribution:** 2
**Rating:** 2
**Confidence:** 3

**Summary:**

Authors propose Counterfacturals and Reasoning for Termination (CaRT), an augmentation plus fine-tuning approach to enable LLMs to balance the explore-exploit tradeoff from verbalized reasoning. CaRT is evaluated on multi-turn interactive medical diagnosis and math problem solving, and shows efficiency over the base model without fine-tuning and the base model fine-tuned on randomly samples training data.

**Strengths:**

The task is interesting and balancing information seeking and reward seeking (or avoid length penalty) is indeed a recurring theme in RL and in reality. The synthesized medical diagnosis domain reflect this tradeoff and is a great testbed.

**Weaknesses:**

1. Using GPT4o to generate termination reasoning as learning data poses a confounder of distillation to your argument, that is, the ability to terminate smartly is distilled from GPT4 and not solely from your "reasoning for termination" construction. Can you use Qwen for termination reasoning?
2. The ablations show that "reasoning for termination" is necessary, but it does not show that a separate termination model/perspective is necessary. I can think of two very simple baselines that can help justify/test the complexity of CaRT.
  * Aggressively greedy system prompt (e.g., "Only ask questions when absolutely necessary")
  * In-house termination reasoning prompt (e.g., each episode/turn, the model itself reasons about termination in context before outputting)
3. Related work such as RL with length penalty (as cited in paper) is excluded from head-to-head comparisons with CaRT without strong empirical evidence for their lack of adaptability.

**Questions:**

See weaknesses.

---

> ### Author Response · Authors · 2025-11-21
>
> Dear Reviewer,
>
> Thank you for the thoughtful and detailed feedback. It seems like the main concerns relate to the use of GPT-4o to generate reasoning traces and the motivation for using a separate termination model. To address these concerns, we have added discussion and new experiments demonstrating the robustness of our method to the reasoning model and joint training. These updates are included in the revised version of the paper, and we summarize the relevant details below. **If our revisions address your concerns, we would be sincerely grateful if you could consider raising your score.**
>
>
> ---
>
> > Using GPT4o to generate termination reasoning as learning data poses a confounder of distillation to your argument ... Can you use Qwen for termination reasoning?
>
> We have added a new experiment to address this concern. We train a variant of CaRT using the base model (Qwen-2.5-3B-Instruct) to generate the termination reasoning traces for the training data in the math setting. We find that using self-generated reasoning maintains good performance, with similar success rates and termination rates as the default CaRT model with GPT-generated reasoning (Figure 7 in **Appendix A**). These results suggest that the efficacy of CaRT does not depend on GPT-4o for generating effective reasoning traces.
>
> ---
>
> > The ablations show that "reasoning for termination" is necessary, but it does not show that a separate termination model/perspective is necessary. I can think of two very simple baselines that can help justify/test the complexity of CaRT.
>
> In our main analysis, we keep the question-asking model separate to focus our evaluation specifically on the termination problem. This choice allows us to avoid the situation in which we observe improved performance of the trained model, but we are unable to distinguish whether the improvement is because the model improved its ability to determine information sufficiency or because the model improved the quality of its questions due to vanilla SFT.
>
> However, we agree that training a joint model for termination and question-asking is an important direction for future work. To address this concern, we have performed a new experiment where we train a single "doctor" model for question-asking, termination, and diagnosis. We compare the CaRT model with the two baselines that you suggest. We find that CaRT demonstrates promising results in this joint setting, achieving similar FRQ success rate and higher Multiple Choice success rate while being more efficient than the baseline reasoning model (Figure 9 in **Appendix C**) and acheiving much higher success rate than the greedy baseline. Additionally, we evaluate diagnosis accuracy - evaluating the model's outputted diagnoses rather than using an external diagnosis model - and find that the CaRT model demonstrates better diagnosis accuracy (+2.2%) than the reasoning base model, again while asking fewer questions.
>
> That said, we note that prior work has addressed the question of "what information to seek" (Li et al. 2025, Zhu et al. 2025) but has not jointly optimized for termination.
>
> > Related work such as RL with length penalty (as cited in paper) is excluded from head-to-head comparisons with CaRT without strong empirical evidence for their lack of adaptability.
>
> To address this concern, we have added a new baseline to our math results which performs RL with a length penalty. We find that this length penalty baseline exhibits poorer termination performance, with lower FRQ SR difference and lower optimal termination rate compared to CaRT and its variants (Figure 7 in **Appendix A**).
>
> ---
>
> ## References
>
> Li, S. S., Mun, J., Brahman, F., Ilgen, J. S., Tsvetkov, Y., & Sap, M. (2025). Aligning llms to ask good questions a case study in clinical reasoning. arXiv preprint arXiv:2502.14860.
>
> Zhu, J., Pan, J., Liu, Y., Liu, F., & Wu, J. (2025, November). Ask patients with patience: Enabling llms for human-centric medical dialogue with grounded reasoning. In Proceedings of the 2025 Conference on Empirical Methods in Natural Language Processing (pp. 2846-2857).

---

> > ### Comment · Reviewer_bJmE · 2025-11-26
> >
> > Thank you for your thoughtful replies. I will raise my score to 4. Not to 6 because it does not feel like the complexity of CaRT is justified through experiment design.
> >
> > ## Use Qwen for termination reasoning
> > This is great. Thank you for the extra experiment.
> >
> > ## Use a single model for termination reasoning and QA
> > Thanks for the new experiments. What's the comparison between CaRT (joint question/termination) in Figure 9 and original CaRT (separate question/termination) on the same eval (the medical one I think) ?
> >
> > If I understand the argument correctly, using separate models for termination and QA arises from the need of controllable experimentation, and not from the model performance wrt. in-context distractions. I do not consider giving a serious attempt to using a single model as future work though, because it is intuitive to implement as a first try and baseline against (let alone massively simplifies the serving stack). It might be hard to get it to work extremely well but nevertheless valuable to be included as baselines and as part of the experiment design to justify your decision of using separate models in CaRT eventually.
> >
> > ## Length penalty
> > My interpretation of Figure 7.1 is that length penalty alone is quite effective at reducing answer length compared to Qwen3-1.7-instruct (length reduced by about 25% at the cost of less than 2% success rate drop). It is indeed worse than full CaRT but I think the better-than-mean performance should grant this baseline a place in your main experimentation (plus a bit more domain specific tuning of alpha).

---

> > > ### Author Response · Authors · 2025-11-26
> > > **Author Response**
> > >
> > > Thank you for your response and further suggestions! We have added additional evaluation, revision, and are working on adding RL baselines to address the final concerns. **Does the new analysis along with the discussion below address your final concern about justifying the separate models in the standard CaRT setting?**
> > >
> > > > Use a single model for termination reasoning and QA
> > >
> > > Yes, our motivation for using a separate termination model is because it allows for controllable experimentation. However, we agree that it is valuable to additionally justify this decision by comparing the efficacy of the standard setup and the joint setup. To address this concern, we have evaluated the joint CaRT model as well as the reviewer's suggested baselines on the holdout evaluation set in our main medical results and updated the corresponding plot (Figure 3a). We have also revised section 5.2 accordingly (purple text) to report the results and better motivate our design choices.
> > >
> > > The overall performance for all the joint models was lower than that of the models and the mean success curve in the setting with separate termination, likely because the questions asked were not as informative. Therefore, these results motivate our decision to train models specializing in question-asking and termination separately. However, we do find that CaRT outperforms the base model variants in this joint setting, achieving even higher FRQ SR than the reasoning model at a lower termination index. These results suggest that CaRT remains effective for improving termination in a setting that has multiple training objectives.
> > >
> > > > length penalty
> > >
> > > Thank you for the suggestion! We are running RL with length penalty for different values of alpha. We will update the main math results with these additional baselines and update you when we are done.

---

> > > > ### Author Response · Authors · 2025-12-01
> > > > **Baseline Update**
> > > >
> > > > We are happy to report that we have finished training RL + length penalty baselines at varying values of alpha (the length penalty coefficient). Across all values of alpha (0, 0.05, 0.1, 0.2), we find that CaRT continues to demonstrate superior termination performance with regard to FRQ difference from mean and optimal termination rate (Figure 7 in Appendix A). Per the reviewers request, we also add this method as a baseline to our main math results (Figure 4), using the value of alpha (0.05) that has the most similar termination rate to CaRT. We hope that these added baselines have addressed the reviewer's final concern.

---

### Official Review · Reviewer_YsUC · 2025-10-30

**Soundness:** 2
**Presentation:** 2
**Contribution:** 2
**Rating:** 4
**Confidence:** 2

**Summary:**

This paper introduces CaRT (Counterfactuals and Reasoning for Termination), a method to teach large language models (LLMs) when to stop information gathering—either by terminating internal reasoning or ending interaction with an environment—in multi-step tasks. The approach combines supervised fine-tuning (SFT) with counterfactual examples and explicit reasoning traces to help the model learn an implicit value function for termination.

**Strengths:**

The paper formalizes and addresses the understudied problem of optimal termination in multi-step LLM reasoning and interaction settings. The combination of counterfactual data generation and explicit reasoning traces is a creative and well-motivated approach to teaching termination. Thorough ablations validate the importance of both counterfactuals and reasoning, and probe how reasoning improves representation generalizability.

**Weaknesses:**

Experiments are confined to two domains (medical diagnosis and math reasoning). Broader evaluation in more diverse or real-world interactive settings is needed.
CaRT only decides when to stop, not what to ask or reason about. This limits its overall impact on end-to-end task efficiency. Could the method be extended to jointly learn both what to ask and when to stop, and what challenges would that introduce?
The RL post-training step does not consistently improve performance and sometimes leads to longer reasoning traces, raising questions about its necessity.
The method relies heavily on accurate external reward labeling (e.g., using Llama-3.1-8B for medical diagnosis), which may not always be available or reliable. Were any experiments conducted with smaller or open-source models for generating reasoning traces, to reduce dependency on GPT-4o.

**Questions:**

See weakness above.

---

> ### Author Response · Authors · 2025-11-21
>
> Dear Reviewer,
>
> Thank you for the thoughtful and detailed feedback. It seems like the main concerns relate to the scope of the method as well as the dependence on other models to create the training data. To address these concerns, we have added discussion and new experiments demonstrating the generalizability of our method. These updates are included in the revised version of the paper, and we summarize the relevant details below. **If our revisions address your concerns, we would be sincerely grateful if you could consider raising your score.**
>
>
> ---
>
> > CaRT only decides when to stop, not what to ask or reason about... Could the method be extended to jointly learn both what to ask and when to stop, and what challenges would that introduce?
>
> In our main analysis, we keep the question-asking model separate to focus our evaluation specifically on the termination problem. This choice allows us to avoid the situation in which we observe improved performance of the trained model, but we are unable to distinguish whether the improvement is because the model improved its ability to determine information sufficiency or because the model improved the quality of its questions due to vanilla SFT.
>
> However, we agree that training a joint model for termination and question-asking is an important direction for future work. To address this concern, we have performed a new experiment where we train a single "doctor" model for question-asking, termination, and diagnosis. We find that CaRT demonstrates promising results in this joint setting, achieving similar FRQ success rate and higher Multiple Choice success rate while being more efficient than a baseline reasoning model (Figure 9 in **Appendix C**). Additionally, we evaluate diagnosis accuracy - evaluating the model's outputted diagnoses rather than using an external diagnosis model - and find that the CaRT model demonstrates better diagnosis accuracy (+2.2%) than the reasoning base model, again while asking fewer questions.
>
> That said, we note that prior work has addressed the question of "what information to seek" (Li et al. 2025, Zhu et al. 2025) but has not jointly optimized for termination.
>
> > The RL post-training step does not consistently improve performance and sometimes leads to longer reasoning traces, raising questions about its necessity.
>
> We agree that RL post-training does not necessarily improve performance, and it is not a component of our main CaRT method. We provide these results as a variation rather than a necessary step. We have revised section 5.1 to clarify this point.
>
> > The method relies heavily on accurate external reward labeling (e.g., using Llama-3.1-8B for medical diagnosis), which may not always be available or reliable.
>
> We emphasize that effective termination is defined relative to the target model's capabilities. Regardless of the model's absolute capabilities, our method provides a principled approach to learn accurate termination signals, by leveraging variation in performance across tasks.
>
> ---
>
> > Were any experiments conducted with smaller or open-source models for generating reasoning traces, to reduce dependency on GPT-4o.
>
> We have added a new experiment to investigate this. We train a variant of CaRT using the base model (Qwen-2.5-3B-Instruct) to generate the termination reasoning traces for the training data in the math setting. We find that using self-generated reasoning maintains good performance, with similar success rates and termination rates as the default CaRT model with GPT-generated reasoning (Figure 7 in **Appendix A**). These results suggest that the efficacy of CaRT does not depend on GPT-4o for generating the reasoning traces.
>
>
> > Experiments are confined to two domains (medical diagnosis and math reasoning).
>
> We believe that broader evaluation in more diverse or real-world interactive settings is an exciting avenue for future work. We are happy to continue the discussion in this direction and would appreciate any suggestions from the reviewer for other domains that we could study.
>
> With regard to the scope of our work, we would like to highlight that we are performing a fairly comphrehensive study, running experiments for the math domain, the in-distribution general medical domain, and an out-of-distribution dermatology medical domain. Previous work on reasoning efficiency is typically confined to the math domain (Muennighoff et al. 2025, Chen et al. 2024, Han et al. 2025). Therefore, we believe that formalizing and tackling the termination problem for both math and medical settings is an important step towards improving efficiency in more interactive settings motivated by real-world challenges.
> ___

---

> > ### Author Response · Authors · 2025-11-21
> >
> > ## References
> > Li, S. S., Mun, J., Brahman, F., Ilgen, J. S., Tsvetkov, Y., & Sap, M. (2025). Aligning llms to ask good questions a case study in clinical reasoning. arXiv preprint arXiv:2502.14860.
> >
> > Zhu, J., Pan, J., Liu, Y., Liu, F., & Wu, J. (2025, November). Ask patients with patience: Enabling llms for human-centric medical dialogue with grounded reasoning. In Proceedings of the 2025 Conference on Empirical Methods in Natural Language Processing (pp. 2846-2857).
> >
> > Muennighoff, N., Yang, Z., Shi, W., Li, X. L., Fei-Fei, L., Hajishirzi, H., ... & Hashimoto, T. B. (2025, November). s1: Simple test-time scaling. In Proceedings of the 2025 Conference on Empirical Methods in Natural Language Processing (pp. 20286-20332).
> >
> > Chen, X., Xu, J., Liang, T., He, Z., Pang, J., Yu, D., ... & Yu, D. (2024). Do not think that much for 2+ 3=? on the overthinking of o1-like llms. Proceedings of the 42nd International Conference on Machine Learning.
> >
> > Han, T., Wang, Z., Fang, C., Zhao, S., Ma, S., & Chen, Z. (2025, July). Token-budget-aware llm reasoning. In Findings of the Association for Computational Linguistics: ACL 2025 (pp. 24842-24855).

---

> > > ### Author Response · Authors · 2025-11-26
> > >
> > > Dear Reviewer,
> > >
> > > We have incorporated the review feedback by running new experiments and revising the paper. We'd really appreciate if you could confirm whether these changes address the concerns about the paper. If we have misunderstood any of the concerns, we'd like to learn so that we can further revise the paper or run additional experiments.
> > >
> > > Best,
> > > The Authors

---

> > > > ### Comment · Reviewer_YsUC · 2025-11-27
> > > > **Response from reviewers**
> > > >
> > > > Thanks for your respoonse. I raise my score.

---

### Official Review · Reviewer_NK21 · 2025-10-31

**Soundness:** 3
**Presentation:** 3
**Contribution:** 3
**Rating:** 4
**Confidence:** 4

**Summary:**

This paper introduces CaRT, a method for teaching LLMs to make strategic termination decisions during multi-step information-gathering tasks. The approach combines counterfactual trajectory pairs with explicit reasoning traces to help models balance task accuracy with resource efficiency. The authors evaluate CaRT on medical diagnosis and mathematical reasoning domains, demonstrating improvements in termination behavior compared to base models and standard fine-tuning baselines.

**Strengths:**

1. The paper addresses an interesting practical challenge: teaching LLMs when to stop gathering information and commit to a decision. This is highly relevant for resource-constrained deployments and agentic applications.
2. The use of counterfactual pairs to isolate critical information is creative and theoretically grounded. Contrasting trajectories where termination is appropriate vs. inappropriate provides a clear learning signal.

**Weaknesses:**

### 1. Model Evaluation
The experiments fine-tune only Qwen2.5-3B-Instruct (for the medical domain). Results from a single small model family are insufficient to support broad claims about the general effectiveness of CaRT. Testing across model sizes or families (e.g., Llama, Mistral) would strengthen the evidence.

### 2. Data
The termination labels may not be stable across models or training stages. The paper uses Llama3.1-8B-Instruct to produce training/evaluation data, while fine-tuning is done on Qwen2.5-3B-Instruct. Models with stronger reasoning ability might succeed with fewer steps, while weaker models need more information. This raises concerns:

* The “golden label” (terminate vs. continue) may shift as base model capability changes. If the underlying reasoning model changes (fine-tuned or uses a different base model for questioning), the termination behavior may also shift, requiring new data generation and training each time.
* When GPT-4o generates rationales for why Llama3.1-8B cannot succeed with certain information, these rationales may not be meaningful: If GPT-4o itself could succeed with the same information, the rationale becomes mere justification rather than true reasoning about information sufficiency or the true uncertainty or capability of the target model.. This raises questions about whether the model learns genuine termination reasoning or simply mimics justification patterns

### 3. Metrics
Several key metrics lack clear explanation:
* Please explain more about FRQ SR Difference from Mean and fixed-budget heuristic baseline
* 'Optimal Termination Rate' methodology is unclear—what does the meaning of the results on data points "excluding conversations without steep increase" (line 301) mean precisely?

The medical domain results in particular seem less convincing—success differences are small and not clearly justified as meaningful.
* Qwen2.5-3B-Instruct + reason prompt achieves the highest success rate (3-4% better than CaRT), though at a higher termination index. While CaRT is more efficient, the natural saturation of benefits from additional information is expected behavior, not necessarily an achievement of the method. Figure 4 shows clear benefits for mathematical reasoning, but the medical diagnosis case lacks compelling evidence for CaRT's advantages beyond natural saturation effects

### 4. Method
* Why are the termination model and diagnosis/reasoning model kept separate? What prevents joint training?
* Were question orders randomized during training? This could significantly impact learning and evaluation.
* The paper compares CaRT with equal-sized SFT datasets. What about SFT with significantly more data (no counterfactuals)?
* How does explicit termination prediction compare with the model's own confidence scores on the target task?

### Minor Issues
* Figure 3 & 4: Mean success rate and Pareto frontier lines are difficult to distinguish visually

**Questions:**

1. How does the approach perform across different model sizes and families?
2. How do you obtain ground truth labels for termination during training, given the dynamic nature of model capabilities?
3. Have you considered joint training of termination and reasoning models?
4. Were question orders randomized during training?
5. How does performance compare when using larger SFT datasets without counterfactuals?

---

> ### Author Response · Authors · 2025-11-21
>
> Dear Reviewer,
>
> Thank you for the thoughtful and detailed feedback. It seems like the main concerns relate to the method's ability to generalize across different base model families/sizes, reward models, and reasoning models. To address these concerns, we have added new experiments showing robustness of our method to these variations. These updates are included in the revised version of the paper, and we summarize the relevant details below. **If our revisions address your concerns, we would be sincerely grateful if you could consider raising your score.**
>
>
> ---
>
>
>
> > The experiments fine-tune only Qwen2.5-3B-Instruct (for the medical domain) ... How does the approach perform across different model sizes and families?
>
> We have added a new ablation experiment investigating the performance of CaRT when applied to Llama-3.1-8B-Instruct and Qwen3-1.7B as base models. Figure 8 in **Appendix B** reports the performance of these CaRT-trained model variants alongside their base models. Across both architectures, CaRT demonstrate superior termination performance compared to their base model counterparts as well as closed-source GPT models. We include a quick summary table of the improvement here.
>
> | Model                       |Improvement in FRQ SR difference| Improvement in Optimal Termination Rate |
> |-----------------------------|-------------------------------|--------------------------------------|
> | CaRT with Qwen3-1.7B   | 4.12%                         | 20.0%                                 |
> | CaRT with Llama3.1-8B-Instruct | 1.39%                         | 23.8%                                 |
>
> ---
>
>
> > The “golden label” (terminate vs. continue) may shift as base model capability changes...  How do you obtain ground truth labels for termination during training, given the dynamic nature of model capabilities?
>
> We have revised section 4 and 5.1 to address this concern (purple text). In all experiments, we use a static diagnosis model (Llama3.1-8B-Instruct) to label each question–answer for success rate, and therefore our termination labels are fixed. Given these termination labels, we then annotate the training samples post-hoc with reasoning about termination. In this setup, the base model’s capabilities do not influence the labels: the reasoning simply helps the model learn a termination policy consistent with a fixed reward model.
>
> We agree that if the diagnosis/reward model itself were to improve, the optimal termination behavior could shift. However, this does not pose a limitation for CaRT. In such settings, CaRT naturally extends to an iterative refinement procedure: as the reward model is updated, new termination labels can be re-generated and the policy can be further fine-tuned. This makes CaRT straightforward to adapt to evolving reward models, and thus more generalizable rather than brittle. Moreover, our experiments across two domains already demonstrate that CaRT learns effective termination behavior for a fixed reward model, supporting its robustness in practice.
>
> ---
>
> > When GPT-4o generates rationales for why Llama3.1-8B cannot succeed with certain information, these rationales may not be meaningful: If GPT-4o itself could succeed with the same information, the rationale becomes mere justification rather than true reasoning about information sufficiency or the true uncertainty or capability of the target model
>
>
> First, we clarify that in our setting, GPT-4o is not asked to judge whether the provided information is sufficient for solving the problem. The termination label (terminate vs. continue) is already fixed by the reward model. GPT-4o is only prompted to produce a short rationale explaining why, given this label, the chosen action (terminate or continue) is appropriate at that point in the reasoning. In other words, GPT-4o is used solely to fill a reasoning gap for supervision—not to determine information sufficiency or to assess the model’s true capabilities. We have revised section 4 to clarify this.
>
> To further address the reviewer’s concern, we add a new experiment in which the termination-reasoning traces are generated by the base model itself (Qwen-2.5-3B-Instruct) for the math setting, instead of using GPT-4o. We find that using self-generated reasoning maintains good performance, with similar success rates and termination rates as the default CaRT model with GPT-generated reasoning (Figure 7 in **Appendix A**).
>
> We also highlight that GPT-4o itself demonstrates poor off-the-shelf termination ability (Figure 8 in Appendix B), indicating that CaRT enables effective termination beyond the models used to generate training data.

---

> > ### Author Response · Authors · 2025-11-21
> >
> > ---
> >
> > > Please explain more about FRQ SR Difference from Mean and fixed-budget heuristic baseline
> >
> > We have revised section 5.1 to clarify these metrics (purple text). To plot the fixed-budget heuristic baseline (Mean Success Rate curve), we perform the following: for each possible termination index, we compute the average FRQ success rate across all the samples in the evaluation set at that particular termination index. This corresponds to the performance of a naive model that always terminates after exactly $n$ interactions/episodes, where $n$ is the termination index.
> >
> > FRQ SR Difference from Mean is the difference between the the FRQ success rate of the evaluated model and the Mean Success Rate curve of this fixed-budget heuristic baseline.
> >
> >
> > > what does the meaning of the results on data points "excluding conversations without steep increase" (line 301) mean precisely?
> >
> > We have revised section 5.1 to be more precise about this filtering step. To compute the optimal termination rate metric for the medical setting, we filter the evaluation set for conversations for which there was an increase in FRQ success rate by at least 50% between two consecutive interactions. Intuitively, optimal termination rate measures: what is the rate at which the model terminates exactly at the point when it acquires very useful information? The other evaluation metrics are still computed on the full evaluation set.
> >
> > ---
> >
> > > Figure 4 shows clear benefits for mathematical reasoning, but the medical diagnosis case lacks compelling evidence for CaRT's advantages beyond natural saturation effects
> >
> > We agree that is difficult to compare CaRT's performance with that of a base model that rarely terminates (due to saturation effects). To compare CaRT with baselines that terminate at a more similar rate, we conducted ablations on the vanilla SFT baseline where we varied the ratio of "terminate" labels to "continue" labels in the dataset. This allowed us to train SFT models that terminated at different rates, mitigating the saturation effect. Figure 5 shows that CaRT outperforms the SFT baseline that terminates at a similar rate, achieving an improvement of ~2.5% in FRQ SR difference and 16% in optimal termination rate.
> >
> > Additionally, we analyze termination rate curves of the base model vs. vanilla SFT vs. CaRT. We find that, while the base-model rarely terminates and the SFT model seems to terminate according to a length heuristic, the CaRT model is able to recognize when sufficient information has been aquired and terminate more appropriately (Figure 6). This analysis suggests that CaRT learns clear signals for termination and demonstrates advantages beyond natural saturation effects.
> >
> > ---
> >
> > > Why are the termination model and diagnosis/reasoning model kept separate? ... Have you considered joint training of termination and reasoning models?
> >
> > In our main analysis, we keep the question-asking model and diagnosis model separate (fixing the diagnosis model) to focus our evaluation specifically on the termination problem. This choice allows us to avoid the situation in which we observe improved performance of the trained model, but we are unable to distinguish whether the improvement is because the model improved its ability to determine information sufficiency or because the model improved its diagnosis ability due to vanilla SFT on correct diagnoses.
> >
> > However, we agree that training a joint model for termination and diagnosis is an important direction for future work. To address this concern, we have performed a new experiment where we train a single "doctor" model for question-asking, termination, and diagnosis. We find that CaRT demonstrates promising results in this joint setting, achieving the highest diagnosis accuracy while being more efficient than a baseline reasoning model (Figure 9 in **Appendix C**).

---

> > > ### Author Response · Authors · 2025-11-21
> > >
> > > ---
> > >
> > > > Were question orders randomized during training?
> > >
> > > Question orders for each conversation were not randomized during training since the termination labels are sequentially dependent on the questions asked up until that point.
> > >
> > > ---
> > >
> > > > The paper compares CaRT with equal-sized SFT datasets. What about SFT with significantly more data (no counterfactuals)?
> > >
> > > To investigate this, we have performed an additional experiment in which we perform vanilla SFT with twice the amount of data. We find that using more data slightly improves the performance of the SFT model but it still does not approach the performance of CaRT (Figure 8 in **Appendix B**). These results suggest that CaRT enables efficient learning of termination that can't be achieved simply by training on more data.
> > >
> > > ---
> > >
> > > > How does explicit termination prediction compare with the model's own confidence scores on the target task?
> > >
> > > We have conducted a new analysis to investigate this. The CaRT model doesn't output confidence scores by default - rather it learns termination signals implicitly. We do train and evaluate a variant of CaRT that verbalizes a confidence score before making the termination decision (CaRT + conf in Figure 5). We analyze this variant for how the termination decision relates to the verbalized confidence score. We find that the model does tend to terminate at a higher rate when verbalizing higher confidence scores (**Appendix D**).
> > >
> > > ---
> > >
> > > > Figure 3 & 4: Mean success rate and Pareto frontier lines are difficult to distinguish visually
> > >
> > > We have increased the saturation of these lines in the figures.

---

> > > > ### Comment · Reviewer_NK21 · 2025-11-25
> > > >
> > > > I thank the authors for their detailed response. I have raised my score.

---

### Author Response · Authors · 2025-12-01
**Discussion Summary**

Dear AC,

We are grateful for the comments provided by the reviewers and are especially grateful for their engagement during the discussion period.

**We would like to highlight that the final comments by Reviewers NK21, YsUC, and bJmE report score increases.** Additionally, the final comment by Reviewer VcE9 states that their concerns were addressed. All of these comments were posted *before* the first report of the OR information leak.

When the reviewing freeze began, we had addressed each reviewer’s concerns, with the exception of two follow-up suggestions from Reviewer bJmE that arose during the discussion period. Although we were unable to continue the conversation, our latest comments on the corresponding discussion thread directly address both of these remaining concerns.

Below, we summarize common concerns raised by the reviewers and explain how we resolved them. We refer the AC to the full discussion thread for detailed replies and clarifications for each individual point.

---

> using GPT-4o to generate reasoning traces for the training data

All reviewers indicated that using GPT-4o to generate reasoning traces was a potential confound in our method. To address this concern, we ran a new experiment in which the termination-reasoning traces are generated by the base model itself (Qwen-2.5-3B-Instruct) for the math setting, instead of using GPT-4o. We find that using self-generated reasoning maintains good performance, with similar success rates and termination rates as the default CaRT model with GPT-generated reasoning (Figure 7 in **Appendix A**). These results suggest that the efficacy of CaRT does not depend on GPT-4o for generating the reasoning traces.

---

> Combining question-asking and termination


Reviewers YsUC, bJmE, and VcE9 suggested that using two seperate models for question-asking and termination in the medical setting was a limitation.

To address this concern, we performed a new experiment where we train a single "doctor" model for question-asking and termination (**Appendix C**). We added evaluation of this joint CaRT model and the joint setting baselines suggested by reviewer bJmE to our main medical results figure (Fig. 3a). The overall performance for all the joint models was lower than that of the models in the standard setting, likely because the questions asked were not as informative. Therefore, these results further motivate our decision to train models specializing in question-asking and termination separately. However, we do find that CaRT outperforms the base model variants in this joint setting, achieving even higher success rate than the reasoning model at a lower termination index. These results suggest that CaRT remains effective for improving termination in a setting that has multiple training objectives.

---

> Additional baselines and ablations

Following the suggestions of Reviewer NK21, we conducted additional ablations over model sizes and model families (Llama-3.1-8B-Instruct and Qwen3-1.7B) for the medical setting. Figure 8 in **Appendix B** reports the performance of these CaRT-trained model variants alongside their base models. Across both architectures, CaRT demonstrates superior termination performance compared to their base model counterparts as well as closed-source GPT models.

Following the suggestion of Reviewer bJmE, we performed additional experiments using RL with length penalty and tuning over the penalty coefficient (Figure 7 in **Appendix A**). Per the reviewer's suggestion, we add this method a baseline to our main math results (Figure 4). We find that CaRT exhibits better termination performance than this baseline, achieving higher FRQ success rate with fewer tokens.

---

Again, we sincerely appreciate the time and care that each reviewer and the ACs have devoted to this process. We hope that the additional experiments, ablations, and baselines have fully addressed all reviewer concerns and improved the paper.

Kind Regards,

The Authors

---

### Meta-Review · Area_Chair_7kJG · 2025-12-24

**Summary:**

The paper addresses the problem that large language models (LLMs) often fail to decide when to stop gathering information and when to act leading to wasted computation or premature decisions. It introduces CaRT (Counterfactuals and Reasoning for Termination) an approach that teaches LLMs optimal termination by contrasting successful trajectories with minimally modified counterfactuals by training the model to explicitly reason about why it should stop or continue. CaRT appears to improve task success on math and medical QA. Overall, this paper is borderline at the end of the review period but I feel that it pushes an important and understudied idea; I think a revised version of this manuscript where the discussions herein are incorporated into the main text of the work would be a significant conference contribution and I encourage the authors to resubmit. I also encourage the authors to explore ways to incorporate qualitative analysis that summarizes the implied stopping criteria learned by the model within the main paper. This would help domain experts quickly assess whether the stopping condition for various Q&A were sensible.

**Reviewer Concerns:**

While adding experiments across additional models improves the work the evaluation is still confined to simulated environments and two tasks. Another concern is whether continuing to iteratively retrained reward models through model evolution is reasonable since it may be costly or impractical in deployment. The work would also benefit from notes or comparisons against other heuristics and proxies for confidence or diminishing returns (e.g. https://arxiv.org/abs/2506.09173 show that one can use expected information gain).

**Reviewer Scores:**

While NK21, YsUC report score increases, bJmE states that moving their score beyond 4 would require further clarity in terms of why the complexity of CaRT is justified through experiment design. While the reviewers do incorporate one experiment in the appendix to address this concern, this paper remains borderline at the end of the review period even taking the increases in score into account.

---

### Decision · Program_Chairs · 2026-01-26

Reject